# On the Role of Hidden States of Modern Hopfield Network in Transformer

**Tsubasa Masumura**\*    **Masato Taki**\*
Graduate School of Artificial Intelligence and Science, Rikkyo University, Japan
{25wr001m,taki_m}@rikkyo.ac.jp

## Abstract

Associative memory models based on Hopfield networks and self-attention based on key-value mechanisms have been popular approaches in the study of memory mechanisms in deep learning. It has been pointed out that the state update rule of the modern Hopfield network (MHN) in the adiabatic approximation is in agreement with the self-attention layer of Transformer. In this paper, we go beyond this approximation and investigate the relationship between MHN and self-attention. Our results show that the correspondence between Hopfield networks and Transformers can be established in a more generalized form by adding a new variable, the hidden state derived from the MHN, to self-attention. This new attention mechanism, modern Hopfield attention (MHA), allows the inheritance of attention scores from the input layer of the Transformer to the output layer, which greatly improves the nature of attention weights. In particular, we show both theoretically and empirically that MHA hidden states significantly improve serious problem of deep Transformers known as rank collapse and token uniformity. We also confirm that MHA can systematically improve accuracy without adding training parameters to the Vision Transformer or GPT. Our results provide a new case in which Hopfield networks can be a useful perspective for improving the Transformer architecture.

## 1   Introduction

The relationship between associative memory in Hopfield networks [22, 1], which has attracted interest from neuroscientists, and Transformers [46] based on key-value memory that have been studied in machine learning has attracted interest from the research community [45, 42, 3, 48, 21]. One of the most interesting results is the finding in [40, 31] that translating modern Hopfield networks into neural networks yields the Transformer architecture that has been very successful in natural language processing [38, 9] and computer vision [12]. What, then, do more general modern Hopfield networks imply for deep learning? This paper gives a concrete answer to this question.

Hopfield networks [22, 1] are a class of models for associative memory. Despite these interesting properties, classical Hopfield networks have the limitation of small storage capacity. Recently, [30] proposed Dense Associative Memory which can achieve storage capacity that scales exponentially or power-wise with respect to the number of neurons by introducing high nonlinearity [7]. These models with large storage capacity are collectively referred to as modern Hopfield networks [40, 31].

Recent advances in Transformer architecture, including its application to language models, have led to significant advances in the study of self-attention mechanisms. These advances have also shed new light on Hopfield networks. In [40], it was shown that the state update rules of modern continuous Hopfield networks (MCHNs) have a mathematical structure exactly equivalent to that of the self-attention mechanism. Furthermore, [31] developed this relationship theoretically, pointing out that

---

\*These authors contributed equally to this work

39th Conference on Neural Information Processing Systems (NeurIPS 2025).

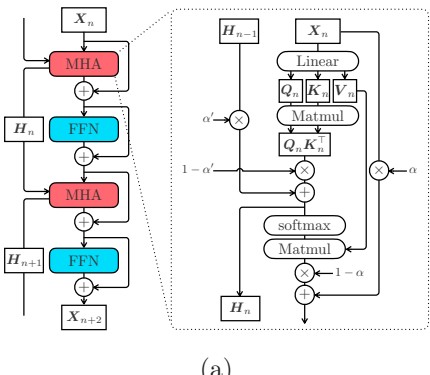
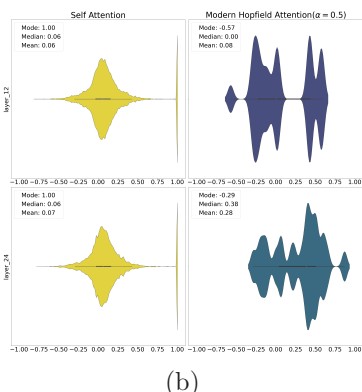

(a)                   (b)

Figure 1: (a) The left figure shows the layer structure of Transformer architecture using modern Hopfield attention (MHA). As the hidden state $H_n$ propagates through each attention layer, information from the upper layer's attention scores is reused in the lower layers. Attention score $Q_n K_n^\top$ is accumulated in the hidden state of each layer, and this value is used for attention calculation. (b) A visualization of the token uniformity in layers 12 and 24 of GPT-2 (Medium) trained on the Wikitext103 dataset, showing a violin plot of the cosine similarity between the tokens. For GPT-2 in the left column, there is a strong peak at similarity 1, and both layers have a mode of 1. On the other hand, in the case of GPT-2 with MHA in the right column, the cosine similarity is kept low and the uniformity of the tokens is dramatically improved.

the self-attention layer of Transformer coincides with the adiabatic limit of generic modern Hopfield networks. Thus, it is expected that there is a deep relationship between the Transformer's architectural design and associative memory. Still the connection, however, lack fundamental understanding.

Therefore, we consider the question: is it possible to interpret modern Hopfield networks without adiabatic approximation in terms of Transformers? The adiabatic limit approximation removes the hidden state dynamics from MCHN. In this paper, we show that maintaining this dynamics introduces a hidden state on the Transformer side, thereby creating a mechanism for the propagation of attention score information from the upper to the lower layers.

By implementing this new attention mechanism into Transformer architectures, this paper introduces a new type of self-attention layer called modern Hopfield attention (**MHA**), as shown in Figure 1(a). MHA does not require additional parameters, and the increase in computational complexity is very small. Nevertheless, simply using MHA instead of the usual attention layer, performance gains can be obtained in various natural language processing and image recognition tasks. Furthermore, we found that MHA effectively solves the problem known as rank collapse, or token uniformity, where Transformer's tokens lose diversity as Figure 1(b). These results indicate that ideas derived from the Hopfield network may provide a new perspective for Transformer research.

In summary, our contributions are as follows:

- By investigating the relationship between MCHN and Transformer beyond the adiabatic approximation, we showed that this correspondence can be further generalized. Based on the correspondence with MCHN, we proposed a new type of attention mechanism with hidden state, MHA. The MHA-based Transformer improves the nature of attention weights by sharing attention score information across layers.

- By training Transformers with MHA, we experimentally showed that the MHA mechanism also contributes to the performance of the Transformers. In particular, we investigated image recognition with Vision Transformer and text generation tasks with GPT-2, and confirmed that MHA actually improves their performance. This method does not generate any additional parameters, and thus can lead to performance gains with only a small increase in computational complexity.

- Theoretically and experimentally, we showed that the reason why MHA works so well as an alternative to self-attention is related to rank collapse. The hidden state of MHA is likely to enhance its performance by cleverly improving Transformer's rank collapse as Figure

1(b). Our results suggest that the Hopfield Networks can provide guidance for improving the Transformer architectures.

## 2   Related Works

The relationship between the modern Hopfield network and the Transformer was investigated in [40, 31], and various improvements and extensions have been made to the modern Hopfield network [34, 53, 26, 25, 41, 5, 21, 24, 23, 49, 19, 17].

In [40], the authors demonstrate the fast convergence of the Modern Hopfield Network (MHN) and justify its use as a conventional module in Transformer-related architectures. On the other hand, in this study's MHA, we propose a dynamic structure that maintains and updates hidden states across layers. More specifically, our MHA naturally incorporates Hopfield recursion into the Transformer layer structure, as "state accumulation and updating" are performed in each layer.

Research on improving the design of Transformer architecture using this relationship has also been conducted in [20]. Unlike these studies, this paper focuses on the effect of keeping hidden state dynamics of MCHN.

In this paper, we saw that hidden states lead to reuse of attention scores across layers. Attention score reuse has been studied [16, 10] from technical perspective, including improvements to the Pre-LN Transformer. These studies are focus only on encoder architecture and do not consider the special combination of moving average with $\alpha'$ and skip connection modification with $\alpha$ as in MHA. On the other hand, this paper showed that the more extended attention mechanisms in MHA can be understood in terms of modern Hopfield networks, and examines its effects including the decoder Transformer. Furthermore, the essential role of MHA is clarified theoretically and experimentally in terms of rank collapse.

## 3   Method: Transformers from Hopfield Network

In this chapter, we review the methods [40, 31] used to derive Transformer from MCHN and give a careful treatment of discretization, which has been ignored in previous studies. As a result, we show that hidden state of MCHN leads to significant changes in the mechanisms of self-attention.

### 3.1   Self-Attention Mechanism

Let $T$ be the number of input tokens with dimension $d$. $\boldsymbol{X}_n \in \mathbb{R}^{T \times d}$ is the feature obtained by concatenating the input token vectors $\boldsymbol{x}_n$ of the $n$-th attention layer. The attention weight of Transformer is given by the row-wise softmax value of the attention score, which is given by the inner product of the query and the key, and the dot-product self-attention is calculated by weighting and adding the value vectors together as $\boldsymbol{X}_{n+1} = \mathrm{softmax}\left(\boldsymbol{Q}_n \boldsymbol{K}_n^\top\right) \boldsymbol{V}_n$, where the query, key, value are given by linear projections of the input $\boldsymbol{X}_n$ as $\boldsymbol{Q}_n = \boldsymbol{X}_n \boldsymbol{W}_Q$, $\boldsymbol{K}_n = \boldsymbol{X}_n \boldsymbol{W}_K$ and $\boldsymbol{V}_n = \boldsymbol{X}_n \boldsymbol{W}_V$ [46]. Each token vector is a slice $\boldsymbol{x}_n = (\boldsymbol{X}_n)_{t,:}$ of the feature tensor. Then the formula for attention mechanism for each token is $\boldsymbol{x}_{n+1} = \mathrm{softmax}\left(\boldsymbol{q}_n \boldsymbol{K}_n^\top\right) \boldsymbol{V}_n$.

### 3.2   Modern Continuous Hopfield Network and its Discretization

MCHN is a network model with bipartite graph connectivity connecting two dynamic variables $\boldsymbol{x}$ and $\boldsymbol{h}$. The connections are given by the network's weights $\boldsymbol{W}$, in which the memories to be associated are stored. $\boldsymbol{x}$ is called the visible state or feature neuron, and $\boldsymbol{h}$ is called the hidden state or memory neuron. In the context of the associative memory model, given a collapsed $\boldsymbol{x}$ as an initial configuration, the complete $\boldsymbol{x}$ is reproduced by association through the time evolution of the state. The time evolution of MCHN is given by the following update rule [31]:

$$\tau_v \frac{d\boldsymbol{x}}{dt} = \boldsymbol{f}\left(\boldsymbol{h}\right) \boldsymbol{W}_1^\top - \boldsymbol{x}, \quad \tau_h \frac{d\boldsymbol{h}}{dt} = \boldsymbol{g}\left(\boldsymbol{x}\right) \boldsymbol{W}_2 - \boldsymbol{h}, \tag{1}$$

where $\tau_{v,h}$ are the time constants of the dynamic system[1]. The activation functions $\boldsymbol{f}(\cdot)$ and $\boldsymbol{g}(\cdot)$ are given by the Lagrangian functions $L_{h,v}$ for $\boldsymbol{h}$ and $\boldsymbol{x}$

$$\boldsymbol{f}(\boldsymbol{h}) = \frac{\partial L_h}{\partial \boldsymbol{h}}, \quad \boldsymbol{g}(\boldsymbol{x}) = \frac{\partial L_v}{\partial \boldsymbol{x}}. \tag{2}$$

In this paper, the vectors $\boldsymbol{h} = (h_a)$, $\boldsymbol{f}(\boldsymbol{h}) = (f_a(\boldsymbol{h}))$, $\boldsymbol{x} = (x_i)$ and $\boldsymbol{g}(\boldsymbol{x}) = (g_i(\boldsymbol{x}))$ are all row vectors. In order to see the correspondence with Transformer below, let us derive discrete time counterpart of MCHN. We then discretize this update rule with a finite difference $\Delta t = t_{n+1} - t_n$ as follows

$$\frac{\tau_v}{\Delta t}(\boldsymbol{x}_{n+1} - \boldsymbol{x}_n) = \boldsymbol{f}(\boldsymbol{h}_n)\boldsymbol{W}_1^\top - \boldsymbol{x}_n, \quad \frac{\tau_h}{\Delta t}(\boldsymbol{h}_{n+1} - \boldsymbol{h}_n) = \boldsymbol{g}(\boldsymbol{x}_n)\boldsymbol{W}_2 - \boldsymbol{h}_n, \tag{3}$$

where $\boldsymbol{x}_n = \boldsymbol{x}(t_n)$ and $\boldsymbol{h}_n = \boldsymbol{h}(t_n)$. Introducing the ratio between the discretization step width and the time constant as $\frac{\Delta t}{\tau_v} = 1 - \alpha$ and $\frac{\Delta t}{\tau_h} = 1 - \alpha'$, we obtain

$$\boldsymbol{x}_{n+1} = \alpha\boldsymbol{x}_n + (1-\alpha)\boldsymbol{f}(\boldsymbol{h}_n)\boldsymbol{W}_1^\top, \quad \boldsymbol{h}_{n+1} = \alpha'\boldsymbol{h}_n + (1-\alpha')\boldsymbol{g}(\boldsymbol{x}_n)\boldsymbol{W}_2. \tag{4}$$

In past studies, the effect of the discretization step $\alpha$ was ignored as negligible, but the precise derivation here leads to an interesting modification of Transformer. In this paper, we give empirical and theoretical results in which $\alpha$ and $\alpha'$'s effect is extremely important.

### 3.3 Adiabatic Limit and Self-Attention

Specific MCHN model is determined by explicitly selecting the Lagrangians. Model B of [31] is given by the following choice of Lagrangians

$$L_h = \log\left(\sum_a e^{h_a}\right), \quad L_v = \frac{1}{2}\|\boldsymbol{x}\|_2^2. \tag{5}$$

These Lagrangians give the activation functions

$$f_a = \mathrm{softmax}(h_a), \quad g_i = x_i. \tag{6}$$

The adiabatic limit in [31] $\tau_h \approx 0$ implies $\boldsymbol{h}_n = \boldsymbol{x}_n\boldsymbol{W}_2$. The update rule for (6) is then given by

$$\boldsymbol{x}_{n+1} = \alpha\boldsymbol{x}_n + (1-\alpha)\mathrm{softmax}(\boldsymbol{x}_n\boldsymbol{W}_2)\boldsymbol{W}_1^\top. \tag{7}$$

Translating (7) by the rule $\boldsymbol{q}_n = \boldsymbol{x}_n\boldsymbol{W}_Q$, $\boldsymbol{W}_1^\top = \boldsymbol{X}_n\boldsymbol{W}_V = \boldsymbol{V}$, $\boldsymbol{W}_2^\top = \boldsymbol{X}_n\boldsymbol{W}_K\boldsymbol{W}_Q^\top = \boldsymbol{K}\boldsymbol{W}_Q^\top$ according to [40, 31], we obtain $\boldsymbol{x}_{n+1} = \alpha\boldsymbol{x}_n + (1-\alpha)\mathrm{softmax}\left(\boldsymbol{q}_n\boldsymbol{K}^\top\right)\boldsymbol{V}$, where $\boldsymbol{X}_n$ is the concatenated tensor of the embedding vectors of all tokens $\boldsymbol{x}_{n1}, \cdots, \boldsymbol{x}_{nT}$. When $\alpha = 0$, i.e., $\Delta t = \tau_v$, this update rule is exactly a usual self-attention mechanism in [46]. In the following, we consider general $\alpha$ and $\alpha'$ to investigate the effect of the hidden state dynamics, which are ignored in the adiabatic limit above.

### 3.4 Hidden State Dynamics and Modern Attention Attention

If the adiabatic limit is not taken and a finite $\frac{\Delta t}{\tau_h}$ is kept, the dynamics of the hidden state is

$$\frac{\tau_h}{\Delta t}(\boldsymbol{h}_{n+1} - \boldsymbol{h}_n) = \boldsymbol{g}(\boldsymbol{x}_{n+1})\boldsymbol{W}_2 - \boldsymbol{h}_{n+1}. \tag{8}$$

In the following, we use the new parameterization $\frac{\Delta t}{\tau_h} = \frac{1-\alpha'}{\alpha'}$ to obtain a simple formula. The dynamics of Model B is then

$$\boldsymbol{x}_{n+1} = \alpha\boldsymbol{x}_n + (1-\alpha)\mathrm{softmax}(\boldsymbol{h}_n)\boldsymbol{W}_1^\top, \quad \boldsymbol{h}_{n+1} = \alpha'\boldsymbol{h}_n + (1-\alpha')\boldsymbol{x}_{n+1}\boldsymbol{W}_2. \tag{9}$$

Using the same translation rules as before, we get the following novel modification of attention layer

$$\boldsymbol{x}_{n+1} = \alpha\boldsymbol{x}_n + (1-\alpha)\mathrm{softmax}(\boldsymbol{h}_n)\boldsymbol{V}_n, \quad \boldsymbol{h}_n = \alpha'\boldsymbol{h}_{n-1} + (1-\alpha')\boldsymbol{q}_n\boldsymbol{K}_n^\top. \tag{10}$$

---

[1]In the following discussion, we do not assume the tying of $\boldsymbol{W}_1$ and $\boldsymbol{W}_2$. This breaking of the symmetry of the memory matrix violates the assumption of monotonically decreasing energy function in the mathematical discussion of [31]. Interpreting the energy function of MHA in asymmetric settings is a very interesting theoretical challenge for future research.

Thus, if the dynamics of the hidden state in the MHN is maintained and mapped to the self-attention layer, a new variable $h$, determined by the value of the attention scores, is added to the self-attention layer. This variable continues to accumulate the value of the attention score in each layer in the form of an exponential moving average across layers. Through this variable, the attention weights of each layer of the Transformer will have a coordinated behavior. In the following, we will investigate the effect of adding this hidden state on the attention layer from the Transformer's perspective. In this paper, this extended attention mechanism with hidden states will be referred to as **M**odern **H**opfield **A**ttention (**MHA**). Compared to the cost $O(dT^2)$ of computing the dot product of self-attention, the computational complexity added by updating the hidden state is about $O(T^2)$. For Transformer that uses more than several hundred dimensions of $d$, this is a small increase in computational complexity.

## 4  Empirical Results

In this chapter, we experimentally investigate how the performance of the model changes when MHA is actually used in place of Transformer's self-attention module. We take the Vision Transformer (ViT) as a representative example of an encoder Transformer model and the GPT-2[39] architecture as a representative example of a decoder Transformer model, and confirm that MHA does indeed lead to systematic performance improvements in several experiments.

### 4.1  Architecture with MHA

In the following, we will focus on the simplest case $\alpha = \alpha'$. It is straightforward to choose both parameter independently, but consider only this case to reduce the hyperparameters. By using our update rule (10) instead of the attention layer, a new tensor called the hidden state $H_\ell$ propagates across the layers. This tensor accumulates the attention score $Q_\ell K_\ell^\top$ in each layer in the form of an exponential moving average. It is not the original attention score that gives the attention weight, but the softmax of the hidden state. At the same time, a skip connection with the weight $(1 - \alpha)$ linked to the coefficient $\alpha$ of the exponential moving average of the hidden state is added according to equation (10), and the balance between the two effects, controlled by $\alpha$, is considered to determine the behavior of the MHA. The detailed structure of the architecture corresponding to (10) is illustrated in Figure 1(a).

In the following experiments, we will employ scaled dot-product attention according to the usual Transformer design and introduce the coefficient $\frac{1}{\sqrt{d_k}}$ in the argument of the softmax function.

### 4.2  Text Generation: GPT-2

To determine the impact of MHA on Transformer performance, we first trained GPT-2 Small(124M) and Medium(350M) [39] on text generation task and tested their performance. The dataset used was WikiText103 [33]. The following experiments in this paper were conducted using up to eight A100 GPUs. The detailed training settings are described in the supplemental material.

To fairly compare the effectiveness of MHA, we trained the GPT-2 architecture and an architecture in which the self-attention layers of GPT-2 are replaced by MHA in the same setting from scratch and compared their perplexity. Table 1 shows the results. The interest of this paper is not to create a SOTA model with detailed hyperparameter tuning, etc., but to see the robustness of the MHA effect, so $\alpha$ was simply set to $0.5$ based on rough hyperparameter search.

As Table 1 shows, there is a clear improvement in perplexity in both the Small and Medium MHA models. Hopfield networks have often been experimented with in comparison to encoder Transformers [40], but our result shows that such comparisons is also useful for decoders.

Table 1: Comparison of the perplexity of GPT-2 and its MHA counterpart trained on the WikiText103 dataset for two cases: GPT-2 Small with 124M parameters and GPT-2 Medium with 350M parameters. In both cases, the introduction of MHA improved the perplexity.

| Small(124M) | | Medium(350M) | |
| --- | --- | --- | --- |
| self-attention | MHA($\alpha = 0.5$) | self-attention | MHA($\alpha = 0.5$) |
| 22.87 | **20.70** | 20.85 | **19.61** |

## 4.3 Text Generation: LLaMA Architecture

To evaluate the effectiveness of Modern Hopfield Attention in more practical text generation architectures, we conducted additional experiments on LLaMA, in addition to GPT-2, using the miniLLaMA implementation. Furthermore, besides WikiText-103, we individually examined cases where CNN DailyMail [18] and BookCorpus [56] were used as training datasets. The results are summarized in Table 2. Even in practical architectures such as LLaMA, whose refined design aims to enhance performance, MHA was found to exert a consistent improvement in perplexity, demonstrating its systematic effectiveness beyond simpler baseline models.

| dataset | self-attention | MHA |
|---|---|---|
| WikiText-103 | 14.49 | **14.29** |
| DailyMail | 19.36 | **18.97** |
| BoocCorpus | 23.76 | **23.50** |

Table 2: Comparison of the perplexity of LLaMA and its MHA counterpart ($\alpha = 0.5$) trained on various datasets. In all cases, the introduction of MHA led to improved perplexity.

## 4.4 Image Recognition: ViT

Next, the Vision Transformer (ViT) was employed as the Transformer decoder model, and again to fairly compare the effect of MHA, two architectures, the ViT architecture and the architecture in which the self-attention layers of ViT are replaced by MHA, were trained in the same configuration. We trained these models in image recognition tasks.

The model used in this study is ViT [12], and the data sets used are CIFAR10/CIFAR100 [29] and ImageNet-1k [8]. The detailed training setup is shown in the supplemental material.

| model size | model type | CIFAR10 | CIFAR100 |
|---|---|---|---|
| ViT-Tiny(5.5M) | self-attention | 93.265 | **73.080** |
| | MHA($\alpha = 0.5$) | 93.015 | 72.030 |
| | MHA($\alpha = 0.7$) | **93.775** | 72.570 |
| ViT-Small(22M) | self-attention | **95.450** | 74.485 |
| | MHA($\alpha = 0.5$) | 95.335 | 75.420 |
| | MHA($\alpha = 0.7$) | 95.440 | **75.590** |
| ViT-Base(86M) | self-attention | 96.190 | 75.360 |
| | MHA($\alpha = 0.5$) | 96.175 | **76.215** |
| | MHA($\alpha = 0.7$) | **96.490** | 75.590 |
| ViT-Large(303M) | self-attention | 96.310 | 72.910 |
| | MHA($\alpha = 0.5$) | 96.500 | **75.775** |
| | MHA($\alpha = 0.7$) | **96.690** | 75.365 |

Table 3: Experimental results are shown for ViTs and their MHA counterparts. For simple tasks such as CIFAR10, performance is close to saturation and there is no clear effect of MHA. On the other hand, for CIFAR100, the performance improvement due to MHA is clear for the larger model. This is a common property of $\alpha = 0.5$ and $\alpha = 0.7$.

### 4.4.1 CIFAR10/100

First, as a simple case, we review the results for CIFAR10 in the left column of the Table 3; for CIFAR10, the effect of MHA is not clearly visible, partly because the performance is basically close to saturation due to the ease of the task. However, it is interesting to note that the effect of MHA is starting to appear in the Base and Large models, which have a high learning capacity. In any case, CIFAR10 is not a sufficient task for the purpose of observing changes in Transformer performance with scratch training.

So let's look at the results for CIFAR100, where the task is more difficult: as shown in Table 3, the larger the model, the larger and clearer the improvement compared to the baseline ViT. Interestingly,

in both cases of the two $\alpha$ choices shown here, the performance improvement relative to ViT can be seen when the model is larger than the Small model.

### 4.4.2 ImageNet-1k

In the experiments on ImageNet-1k, due to computational resource constraints, we adopt ViT-B (86M) as a model of good enough size to obtain nontrivial training results. The results of 300-epoch training of ViT-B and its MHA counterpart from scratch with ImageNet-1k are shown in Table 4. Following the standard training setup, AdamW[32] was used for optimizer and cosine decay for learning rate scheduling. Random erasing[54], mixup[52], cutmix[51], and RandomAugment[6] were used for augmentation. For details, please refer to the supplemental material.

Table 4: Classification validation accuracies for ViT-B(86M) and its MHA counterpart.

| Data set | self-attention | MHA($\alpha = 0.5$) | MHA($\alpha = 0.7$) |
|---|---|---|---|
| ImageNet-1k | 76.074 | 76.434 | **77.058** |

As shown in Table 4, the performance improvement in ViT-B was also observed in ImageNet-1k. Although the performance improvement is less than 1%, this performance difference is considered a non-trivial result compared to the examples in previous studies on ViT improvement. As in previous experiments, this increase in performance is produced by adding only a small amount of computation without adding any training parameters. This is an interesting result, which suggests that hidden states may help improve attention mechanisms. In the next chapter, we will investigate both theoretically and experimentally regarding how hidden states produce these performance gains.

### 4.4.3 Downstream Tasks

To evaluate MHA's effectiveness across diverse tasks, we measured the transfer performance of a pre-trained ImageNet model using linear probing. Using a pre-trained ViT and its MHA counterpart as backbones, we conducted transfer learning experiments on four commonly used downstream datasets (Oxford Flowers 102 [35], Food-101 [4], Stanford Dogs [27], and Stanford Cars [28]). Results in Table 5 show that the MHA variants achieve consistently good transfer performance.

| dataset | self-attention | MHA |
|---|---|---|
| Flower102 | 81.15 | **93.85** |
| Food101 | 74.51 | **87.99** |
| Stanford dogs | **95.00** | 83.64 |
| Stanford cars | 51.54 | **87.54** |

Table 5: Comparison of the transfer accuracy of ImageNet-1K pre-trained ViT and its MHA counterpart ($\alpha = 0.7$) on various downstream datasets.

### 4.5 Effect of Combining $\alpha$ and $\alpha'$

Our update rule (10) has two hyperparameters $\alpha$ and $\alpha'$, but for simplicity, we have so far restricted our discussion to the case where both values are equal $\alpha = \alpha'$. However, as shown in Figure 1(a), these two quantities essentially work differently. $\alpha$ is a quantity that balances the value after attention computation and the strength of the skip connections in the attention module. On the other hand, $\alpha'$ is the coefficient of the exponential moving average in accumulating the attention scores to hidden states.

A nontrivial result (10) derived from MCHN is that these two independent effects are simultaneously added to the Transformer. To see whether these two are really both necessary, or whether they work in concert, let us try an experiment in which $\alpha$ and $\alpha'$ are varied independently.

Table 6 shows the change in performance when one of the hyperparameters is fixed at 0.5 and the value of the other is varied. When only $\alpha$ is moved from the original $\alpha = 0.5 = \alpha'$ to $\alpha = 1$, the performance drops to chance-level accuracy. This is evident from the fact that all values except for the skip connection are set to 0. On the other hand, when $\alpha$ is set to 0, the performance degrades to

Table 6: Performance change of ViT-T when two hyperparameters are changed independently

| CAIFAR100 MHA($\alpha = 0.5$) | | | | | | | | | | | |
|---|---|---|---|---|---|---|---|---|---|---|---|
| $\alpha'$ | 0.0 | 0.1 | 0.2 | 0.3 | 0.4 | 0.5 | 0.6 | 0.7 | 0.8 | 0.9 | 1.0 |
| score | 71.16 | 71.13 | 72.29 | 70.77 | 72.12 | 72.13 | 72.06 | 72.02 | 71.98 | 70.72 | 66.10 |
| CAIFAR100 MHA($\alpha' = 0.5$) | | | | | | | | | | | |
| $\alpha$ | 0.0 | 0.1 | 0.2 | 0.3 | 0.4 | 0.5 | 0.6 | 0.7 | 0.8 | 0.9 | 1.0 |
| score | 69.89 | 71.20 | 71.26 | 71.64 | 72.02 | 72.13 | 72.66 | 70.52 | 70.46 | 67.70 | 1.00 |

69.89. Thus, it can be seen that further performance improvement is realized by adding not only $\alpha'$ but also $\alpha$. Similarly, when $\alpha$ is fixed, setting $\alpha'$ to 0 also results in poorer performance.

## 5 How MHA improves Transformers

### 5.1 Problem and Improvement of Transformer Layers

Next, let us examine why MHA leads to performance gains in various tasks. It is known that as the depth of Transformer increases, training becomes more difficult and performance tends to saturate rapidly. The phenomenon of rank collapse has been discussed as one cause of this problem. It is possible that our model mitigates the problem without explicit regularization or other means. Therefore, we provide below some theoretical and empirical results that support this hypothesis.

#### 5.1.1 Rank Collapse

It has been observed that as the depth of the Vision Transformer increases, the patch tokens become extremely similar and rapidly lose diversity [44][55]. This phenomenon is now understood as token uniformity, rank collapse, or oversmoothing [11, 44, 55, 50, 36, 43, 15, 14, 37, 47, 2, 13][2]. Various innovations have been proposed to reduce this problematic phenomenon in order to improve Transformer performance [44, 55, 2].

Rank collapse [11] is defined as a phenomenon in which Transformer feature rapidly collapses into a rank 1 matrix with increasing depth. Thus, for the feature $\boldsymbol{X}^{(L)}$ of the $L$-th layer, rank collapse is formulated as the property that the residual of deep Transformer feature rapidly converges to zero as follows

$$\|\text{Res}(\boldsymbol{X}^{(L)})\| \approx 0 \text{ for } L \gg 1, \tag{11}$$

where $\text{Res}(\boldsymbol{X})$ is the residual $\text{Res}(\boldsymbol{X}) = \boldsymbol{X} - \mathbf{1}\boldsymbol{x}^\top$ for $\boldsymbol{x} = \arg\min_{\boldsymbol{x}} \|\boldsymbol{X} - \mathbf{1}\boldsymbol{x}^\top\|$. This convergence means the feature is approximately a rank one matrix $\boldsymbol{X}^{(L)} \approx \mathbf{1}\boldsymbol{x}^\top$.

### 5.2 Theoretical Implication

For a clear theoretical analysis of the causes of rank collapse, we consider a deep network consisting of only the self-attention layers according to [11]. It is also straightforward to extend the discussion to the actual Transformer architecture [11]. Let us consider a self-attention-only network consisting of $L$ layers without skip connection

$$\text{AttnNet}(\boldsymbol{X}) = \text{MHSA} \circ \cdots \circ \text{MHSA}(\boldsymbol{X}). \tag{12}$$

$\text{MHSA}(\boldsymbol{X})$ is the multi-head self-attention module. The number of heads and embedding dimension of each MHSA are $H$ and $d_k$. In [11], this attention-only network has been shown to cause very serious rank collapse:

**Theorem 5.1** ([11]). *The norm of the residual of attention-only network AttnNet($\boldsymbol{X}$) decays as*

$$\|Res\left(AttnNet(\boldsymbol{X})\right)\|_{1,\infty} \leq (rC)^{\frac{3^L-1}{2}} \|Res\left(\boldsymbol{X}\right)\|_{1,\infty}^{3^L}, \tag{13}$$

*where $r = \frac{8H}{\sqrt{d_k}}$ and $C$ is certain constant. This suggests the double exponential decay of the rank.*

---

[2]The rank collapse in [11] refers to the phenomenon where the tokens corresponding to each row of a feature become perfectly proportional vectors. This means perfect token uniformity. On the other hand, the phenomenon observed in actual Transformers is that many, if not all, tokens are perfectly aligned, forming a group of tokens with a mutual cosine similarity of 1.

The definition of the norm $\| \cdot \|_{1,\infty}$ in this paper is the composite of operator norms $\|\boldsymbol{X}\|_{1,\infty} = \sqrt{\|\boldsymbol{X}\|_1 \|\boldsymbol{X}\|_\infty}$.

In [11], it was shown that skip connection and the addition of an FFN layer are effective in reducing this serious collapse. Interestingly, however, even though our MHA is not specifically designed to prevent rank collapse, it is able to prevent the decay phenomenon in attention-only networks without any skip connection. Even when removing skip connections completely by setting $\alpha = 0$, a non-zero $\alpha'$ leads to the following mitigation of rank collapse in the attention-only network:

**Theorem 5.2.** *By keeping non-zero $\alpha'$, the upper-bound of inequality evaluation is improved as follows*

$$\|Res\big(AttnNet(\boldsymbol{X})\big)\|_{1,\infty} \leq \max{}_{m=0}^{L}\, (r(1-\alpha')C_1)^{\frac{3^m-1}{2}}\, (r\alpha'C_2)^{3^m(L-m)} \|Res\big(\boldsymbol{X}\big)\|_{1,\infty}^{3^m}. \quad (14)$$

*This suggests the avoidance of exponential decay.*

*Proof.* See the supplemental material for detailed proof. The sketch of the proof is as follows: by introducing the hidden state as $\alpha' \neq 0$, the decaying effect of rank by single attention layer can be evaluated as follows

$$\|\mathrm{Res}\big(\mathrm{MHSA}(\boldsymbol{X})\big)\| \leq \max\big(r_1(1-\alpha')\|\mathrm{Res}\big(\boldsymbol{X}\big)\|^3,\, r_2\alpha'\|\mathrm{Res}\big(\boldsymbol{X}\big)\|\big), \quad (15)$$

where $r_{1,2} = rC_{1,2}$ and the norm here is $\| \cdot \|_{1,\infty}$. Notice that the second argument in the max function significantly reduces the third-order decaying effect in [11]. By applying this inequality repeatedly over $L$ layers, we obtain the following inequality

$$\|\mathrm{Res}\big(\mathrm{AttnNet}(\boldsymbol{X})\big)\| \leq \max\left( (r_1(1-\alpha'))^{\frac{3^L-1}{2}} \|\mathrm{Res}\big(\boldsymbol{X}\big)\|^{3^L}, \cdots, (r_2\alpha')^L \|\mathrm{Res}\big(\boldsymbol{X}\big)\| \right), \quad (16)$$

where $\cdots$ means $(r_1(1-\alpha'))^{(3^m-1)/2}(r_2\alpha')^{3^m(L-m)}(\|\mathrm{Res}(\boldsymbol{X})\|)^{3^m}$ for $m = 1, \cdots, L-1$. $\quad\square$

On the right hand side of this inequality (14), the $m = L$ term is the very term that created the double exponential decay of the original self-attention mechanism [11], but the $m = 0$ term dominates in (14) and relaxes the rank decay to linear decay as $(r\alpha'C_2)^L \|\mathrm{Res}\big(\boldsymbol{X}\big)\|_{1,\infty}$ since

$$(r_1(1-\alpha'))^{(3^m-1)/2}(r_2\alpha')^{3^m(L-m)}(\|\mathrm{Res}(\boldsymbol{X})\|)^{3^m} < (r_2\alpha')^L(\|\mathrm{Res}(\boldsymbol{X})\|)^L. \quad (17)$$

Note that we assume $r_{1,2}, \|\mathrm{Res}(\boldsymbol{X})\| < 1$ following the logic of [11]. This decaying factor is controlled by the hidden states of the $h$-th head of the $\ell$-th layer $\boldsymbol{H}_{\ell,h} = \alpha' \boldsymbol{H}_{\ell-1,h} + (1-\alpha')\boldsymbol{Q}_{\ell,h}\boldsymbol{K}_{\ell,h}^\top$ and the weight matrix $\boldsymbol{W}_{VO,h}^{(\ell)}$ for the value and output linear projection of attention module as $C_2 = \max_\ell \max_h \|\boldsymbol{W}_{VO,h}^{(\ell)}\|_{1,\infty}\|\boldsymbol{H}_{\ell,h}\|_1$.

In [11], such an effect was created by introducing skip connection, but in the MHA, the hidden state contribution already produces such an effect without using skip connection. Also, setting $\alpha' = 0$ reproduces the double exponential decay results of the original attention-only network (13).

## 5.3 Empirical Results

Using the theoretical analysis setup used in previous studies, we showed that MHA can effectively prevent rank collapse in the previous section. However, since these setups are based on several theoretical simplifications, it is unclear whether the rank collapse reduction also occurs in actual Transformers. In particular, it is not clear whether the introduction of MHA has any further effect in usual architectures with skip connection to reduce rank collapse. In this section, we will confirm that MHA does indeed further reduce rank collapse in a few controlled experiments.

Since the skip-free network was shown to suffer from rank collapse as it gets deeper, let's examine the effect of MHA on the actual performance degradation with depth. Table 7 shows the results of trained models from depths 1 to 12 for the skip-free networks and their MHA versions, and evaluating their performance. As can be seen from the results in the left column of the table, when the depth increases beyond 4 layers, the performance drops sharply due to multilayering. On the other hand, for the models in the right column using MHA, it can be seen that the degradation of the model due to multilayering is kept at a fairly mild level. Thus, the MHA model can effectively utilize the depth of

Table 7: Changes in performance as skip-free networks based on ViT-T are deepened.

| depth | self-attention | | MHA | |
|---|---|---|---|---|
| | CIFAR10 | CIFAR100 | CIFAR10 | CIFAR100 |
| 1 | 55.08 | 30.90 | 65.41 | 40.08 |
| 2 | 63.72 ↑ | 40.06 ↑ | 79.75 ↑ | 56.94 ↑ |
| 4 | 57.38 ↓ | 32.25 ↓ | 85.74 ↑ | 64.39 ↑ |
| 8 | 48.59 ↓ | 17.19 ↓ | 80.34 ↓ | 49.90 ↓ |
| 12 | 10.00 ↓ | 1.00 ↓ | 10.00 ↓ | 1.00 ↓ |

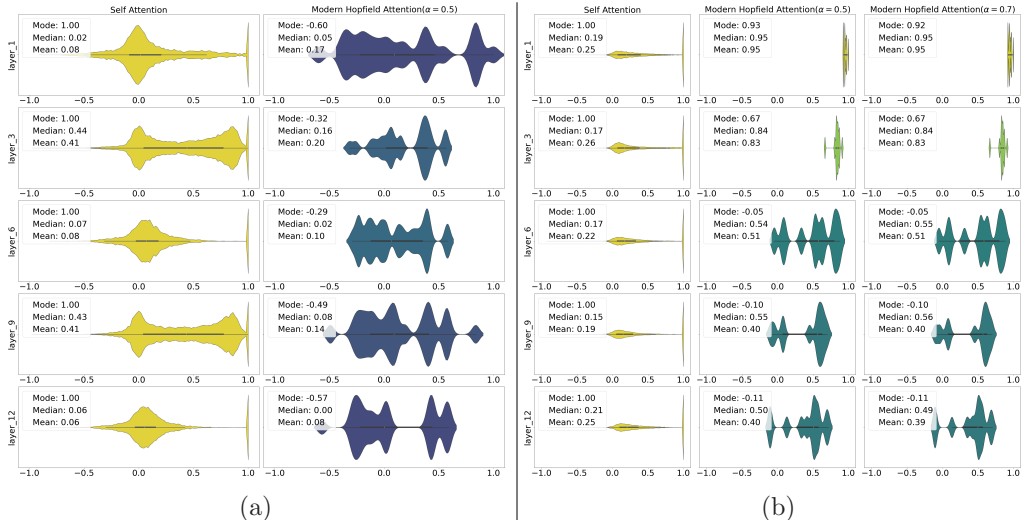

Figure 2: The violin plots of cosine similarity between tokens in several layers for (a) GPT-2 (Medium) trained on Wikitext103 and (b) ViT-B trained on CIFAR100. MHA layers with high average similarity of tokens exist, but tokens with a perfect similarity of 1, as in the case of self-attention, disappear, preventing their ranks from dropping.

the model than the original model. Therefore, it is highly likely that MHA can significantly improve rank collapse, which becomes more severe as the network becomes deeper, even in real networks.

Next, let us examine cases of actual Transformer architectures with skip connections and FFN layers. Figure 2 shows the measured cosine similarity between tokens in several layers for GPT-2 (Medium) and ViT-B trained on CIFAR100, displayed as violin plots. See the supplementary material for more detailed plots. It is noteworthy that in the cases of normal GPT-2 and ViT-B, the mode of similarity is 1.0 for all layers, while the violin plot shows a sharp peak around 1.0. This indicates that even with the addition of the skip connection and FFN layers, there is still a non-negligible token uniformity, or partial rank collapse. On the other hand, the results for the GPT-2 and ViT models with MHA show that the peaks in the original models have disappeared and the mode values have been reduced to very small values. This indicates that MHA does indeed play a role in dramatically removing token uniformity in GPT-2 and ViT.

## 6    Conclusion

In this paper, we examine the question of whether new insights can be obtained from the modern Hopfield network for Transformer. The results showed that by introducing the hidden state of MCHN into Transformer, a new attention mechanism called MHA, which inherits attention scores from layer to layer, has been discovered and can be useful for improving ViT and GPT performance. MHA was also found to play a role in solving the rank collapse problem in deep Transformer. The MHA's mechanism to prevent the rank collapse may have contributed to Transformer's improved performance. We hope that this research will open new possibilities for the systematic design of Transformer architectures using Hopfield networks.

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
