# OpenReview forum: "On the Role of Hidden States of Modern Hopfield Network in Transformer"
_NeurIPS.cc/2025/Conference — NeurIPS 2025 poster_

### Official Review · Reviewer_BWoL · 2025-06-15

**Clarity:** 3
**Significance:** 4
**Originality:** 4
**Rating:** 4
**Confidence:** 4

**Summary:**

Recently, it was shown that the state update rules of modern continuous Hopfield networks (with softmax-based energy) have a mathematical structure exactly equivalent to that of the self-attention. Building upon this connection, this work shows that the connection between the two systems can be more generalized, in which the authors proposed a new attention mechanism with the hidden state, Modern Hopfield Attention (MHA). The MHA-based Transformer improves the nature of attention weights by sharing attention score information across layers.

**Questions:**

1. Regarding the inverse temperature $\beta$, did you simply use the standard value of $\frac{1}{\sqrt{d}}$ as $\beta$? It is worth including $\beta$ as part of your formulations detailed in section 2.3 for completeness.

2. For GPT-2 (medium), MHA layers exhibit "softer" cosine similarity. But for ViT or image domain, this does not seem to be the case for the early layers. Why is that for the early layers the cosine similarity centers around 1, but not for the later layers in the ViT/Image domain?

3. In some sense, I can see that there are two energies operating independently: one for the MLP block and the other for the attention (which is MHN). The hidden state coming from MLP block is evolving following its own energy, and the same applies to the attention. Overall, what is your intuition of this aspect?

**Ethical Concerns:**

["NO or VERY MINOR ethics concerns only"]

**Final Justification:**

I think the work is good and the presented results are significant. My overall score of 4s imply reflects that they need to demonstrate more results, e.g., showing the differences in self-attention and MHA regarding the learned embeddings and text modality. **I trust that the authors will include those final results in their last version.**

**Limitations:**

There is not a single global energy function controlling the operations of the Transformer like [Hoover et al., (2023)](https://proceedings.neurips.cc/paper_files/paper/2023/file/57a9b97477b67936298489e3c1417b0a-Paper-Conference.pdf) --- making the model in this work, not a recurrent model. Nonetheless, I think the formulation, presented here, is interesting and novel as it enables exponential moving average of the hidden states (from MLP) and tokens (from attention).

**Paper Formatting Concerns:**

In your appendix, there are figures which illustrate a lot of violin plots. I think it is unnecessary to show too many as it is difficult to see them all --- you should try to slim it down.

**Quality:**

4

**Strengths And Weaknesses:**

Strengths:

1. Unlike [Hoover et al., (2023)](https://proceedings.neurips.cc/paper_files/paper/2023/file/57a9b97477b67936298489e3c1417b0a-Paper-Conference.pdf), this work does not require training but rather reformulating the operation of Attention to match MHN update and to incorporate the input of the MLP output in Transformer into the MHN update. Hence, there is no parallel update of the token between MLP and Attention as in [Hoover et al., (2023)](https://proceedings.neurips.cc/paper_files/paper/2023/file/57a9b97477b67936298489e3c1417b0a-Paper-Conference.pdf).

2. The Transformer operation, presented by the work, is now essentially an exponential moving average of the MLP outputs and Attention outputs across a series of layers, which could indicate a more robust representation of the tokens after they are updated.

3. The results illustrate strong performance of the modified Transformer as the model's size becomes larger, see Table 2. Moreover and most importantly, the authors provide an understanding on why the introduced method has a better performance and highlighted the problem of rank collapse in Transformer --- cool!

Weaknesses:

1. The results demonstrated on the text modality are little. It is still unclear if the same trend is applicable for the language modality and multi-modalities domain. It would be nice to see further demonstration of the introduced approach on text, see [He et al. (2024)](https://arxiv.org/pdf/2402.13449).

2. In the MCHN language, both of the key and value matrices, $W_K$ and $W_V$, are treated as the memory matrix $\xi$, i.e., the memory matrix is shared between the two operations. But in this work, they are treated separately --- following the same line as the standard Transformer operation.

3. The work reformulates the Transformer's operations via the perspective of MHN dynamics, but it is not clear on how these new operations follow a global energy function (which dictates the MHN dynamics). The work should try to highlight and explain this aspect better, since the usage of a well-formulated global energy function is one of the fundamentals of Associative Memory.

---

> ### Author Rebuttal · Authors · 2025-07-31
>
> We sincerely appreciate your valuable review. In particular, we appreciate your positive evaluation of the novelty of this study, its significance in the context of Hopfield Networks, and its theoretical contributions.
>
> **[W1] Verification of effectiveness in text modality**
>
> As you pointed out, strengthening performance verification in natural language is an important issue. We conducted additional experiments using the Llama architecture (mini-llama implementation) to verify the effectiveness of MHA in other Transformer-based architectures. We trained the model on 20% subsets of Wikitext103, BookCorpus, CNN/DailyMail, and XSum, and confirmed significant improvements in perplexity as follows:
>
> **Additional results in Llama (Perplexity)**
> | Dataset |  Self-attention | MHA($\alpha=0.5$) |
> | ---- | ---- | ---- |
> | Wikitext103(20%)| 29.92| **24.48** |
> | BookCorpus(20%)| 67.05 | **40.68** |
> | CNN_dailymail(20%)| 78.81| **58.62** |
> | xsum(20%)| 103.30| **97.91** |
>
> These results demonstrate that MHA is consistently effective across multiple data domains, reinforcing its usefulness in natural language processing. On the other hand, verifying its effectiveness in multimodal domains is an important future direction.
>
> **[W2, W3, Q3] Considerations on the symmetry of memory matrices and energy functions**
>
> Thank you for your important questions regarding key matrices and value matrices. As you pointed out, in MCHN, the weight matrix connecting the hidden state and the visible state is given by the same memory matrix $\xi$ and $\xi^\top$. On the other hand, in attempts to relate MCHN to Transformers, such as [35][27], the weight tying of $\xi$ and $\xi^\top$ is resolved, and they are mapped to independent $W_K$ and $W_V$, respectively.
>
> The violation of symmetry in this memory matrix destroys the assumption of monotonic decrease in the energy function in the mathematical argument of Krotov-Hopfield [27]. Therefore, in such cases, it becomes impossible to prove the existence of a monotonic decrease in the energy function. On the other hand, as a trade-off, we can obtain improved representation and learning capabilities in Transformer models. For example, in [KOT], this issue is investigated in the context of the correspondence between Krotov-Hopfield's MCHN and MLP-Mixer. According to this study, it has been confirmed that breaking symmetry significantly improves image recognition performance.
>
> Furthermore, according to [YM], breaking the symmetry of the memory matrix may cause bifurcations in energy dynamics. The energy function interpretation of MHA in an asymmetric setting of $\xi$ is an extremely interesting theoretical issue for future research. As you pointed out, the fact that there are unresolved and interesting issues regarding the energy function is extremely important, so we will clearly state this in the camera-ready manuscript.
>
> **references**
> - [KOT] Karakida-Ota-Taki, Hierarchical Associative Memory, Parallelized MLP-Mixer, and Symmetry Breaking
> - [YM]Yampolskaya-Mehta, Controlling the bifurcations of attractors in modern Hopfield networks, NeurIPS 2023 Workshop AMHN
>
>
> **[Q1] Selection of inverse temperature**
>
> As you pointed out, in this study, we selected the inverse temperature as $\beta=\frac{1}{\sqrt{d_k}}$. In the equations in Section 2, we omitted the inverse temperature for the sake of simplicity. We briefly mentioned this point in line 146, but it was insufficient in terms of clarity for readers. In the camera-ready version, we will revise Section 2 to explicitly state this setting.
>
> **[Q2] Behavior of cosine similarity in the initial layer**
>
> Thank you for your interesting comment. In Figure 2, the cosine similarity of a specific layer indicates the cosine similarity between tokens input to that layer. Therefore, the cosine similarity of the first layer is the cosine similarity between the embedding vectors created in the embedding layer. The nature of embedding differs greatly between natural language and images.
>
> In particular, in the initial layer, while tokens have discrete meanings in natural language, in the image domain, embeddings for each patch have spatial continuity, and the similarity between adjacent image tokens tends to be high. This explains the difference in distribution in the initial layer of ViT.
>
> However, this difference in the initial distribution specific to each domain gradually diminishes through MHA's multi-layer processing, and ultimately both converge to a state where token uniformity is improved. We consider this observation to be supporting evidence that MHA consistently performs conversion of representation even across different domains.
>
> **[Comments on formatting]**
>
> Thank you for your feedback. As you pointed out, the number of violin plots in the current appendix is excessive and may hinder interpretation. The purpose of these plots was to comprehensively show how the similarity distribution of each token changes across layers and architectures. In the camera-ready version, we will revise the appendix to simplify these plots.
>
> We appreciate the opportunity to reconsider the theoretical background and future possibilities of our method through the review comments.

---

> > ### Comment · Reviewer_BWoL · 2025-08-02
> > **Thank you for the response and additional experiments.**
> >
> > ## TLDR
> > With the new results on text modality, I think it's fair to say that the introduced approach is performative in the text domain. However, I highly recommend that for the final print --- the authors should use the entire dataset instead of 20% of it. To me, there is one question regarding this aspect: does self-attention require more data to perform better? Or, indeed the introduced method does perform better even when there are more data.
> >
> > ## Questions/Suggestions
> > 1. I must have missed this aspect before. In your table 4, for the row with CIFAR100, when $\alpha = 1$, your score is 1.0. Is this a mistake? I expect the score to decrease but this is a drastic drop off from 67.70 (at $\alpha = 0.9$) to 1.0
> >
> > 2. How do the embeddings of your learned matrices in your attention differ from self-attention? I think you should include that aspect.
> >
> > 3. So, it is still quite unclear on the exact energy which governs these introduced dynamics. Certainly, we can see the benefits of having separate dynamics (between FFN and MHA) here. Hence, please do discuss this more in-depth in your final version.
> >
> > 4.. I think Fig. 1 quality is bad. It needs some improvement for the final print. To my eyes, it does feel pixelated, especially the texts (e.g., Linear, Matmul, etc.).
> >
> > ## Overall Feeling
> > I am satisfied with the response but I will need some time to reflect. I hope the authors can continue answering my questions.

---

> > > ### Comment · Reviewer_aW1E · 2025-08-03
> > > **Complementary Comment**
> > >
> > > I would like endorse Reviewer BWoL's comment here that although the experiments for text modality are significant and much appreciated, especially given the limited time the authors had, it is strongly encouraged to report results on the whole dataset in the camera-ready version of the paper, if it gets accepted at this conference.

---

> > > > ### Author Response · Authors · 2025-08-06
> > > >
> > > > Thank you very much for your positive evaluation and valuable suggestion. We fully agree that evaluating on the full dataset is important to enhance the reliability and applicability. We are currently securing computational resources, and we plan to obtain and report the full-dataset results in the camera-ready version of the paper.

---

> > > > > ### Comment · Reviewer_BWoL · 2025-08-06
> > > > > **Good Responses**
> > > > >
> > > > > To the authors,
> > > > >
> > > > > Thank you for your detailed response.
> > > > >
> > > > > Regarding your new result on the cosine similarity statistics for MHA with $\alpha = 0.7$, I frankly do not know how to interpret these results, without knowing the full details of the experiments. But, if indeed you are comparing embeddings of different inputs, these results indicate that different embeddings are orthogonal to each other. This is good.
> > > > >
> > > > > **Overall, I will raise my score on significance from a 3 to 4.** I think the work presents good results. I am more on the line of maintaining my overall score of 4 (but I will spend more time thinking about this aspect). Frankly, I do not think the work is bad, but I would love for the authors to incorporate the results on text datasets (to the full extent). Thus, once again, my overall score of 4 simply reflects the need for those results and not that the paper is slightly flawed.

---

> > > > > > ### Author Response · Authors · 2025-08-08
> > > > > >
> > > > > > Thank you very much for your constructive feedback.
> > > > > >
> > > > > > Regarding the new results on cosine similarity statistics for MHA, these were obtained by inputting the ImageNet-1k validation dataset into ViT-B model trained on ImageNet-1k with MHA ($\alpha=0.7$). The statistics are similar to those accompanying the violin plots shown in Figures 1(b) and 2 of the paper.
> > > > > >
> > > > > > We sincerely appreciate your positive evaluation reflected in raising the significance score from 3 to 4.
> > > > > >
> > > > > > As you kindly pointed out, we are currently working on adding comprehensive results for text datasets, which we plan to include in the camera-ready version. We believe these additions will further strengthen the paper.
> > > > > >
> > > > > > Once again, thank you very much for your thoughtful and encouraging comments.

---

> > > ### Author Response · Authors · 2025-08-06
> > >
> > > Thank you again for taking the time to provide your thoughtful questions and suggestions.
> > >
> > > **Regarding validation using the entire dataset and the scalability of the method**
> > > >However, I highly recommend that for the final print --- the authors should use the entire dataset instead of 20% of it.
> > >
> > > Thank you for your suggestion. We fully agree with its importance. We are currently securing computational resources to conduct experiments using the full dataset, and we plan to include the updated results in the camera-ready version.
> > >
> > > >To me, there is one question regarding this aspect: does self-attention require more data to perform better? Or, indeed the introduced method does perform better even when there are more data.
> > >
> > > This is indeed a very insightful and fundamental question. At present, we do not have clear empirical results on large-scale data, so we cannot draw definitive conclusions. However, from a theoretical perspective, this method has a structure that addresses the rank collapse problem inherent to the Transformer, which we believe is a universal issue independent of data size. Therefore, we believe that MHA can still contribute to improvement even as data size increases. This is an important research topic that requires further verification through larger-scale experiments.
> > >
> > > **Regarding the "1.0" value for CIFAR100 in Table 4**
> > > > How do the embeddings of your learned matrices in your attention differ from self-attention? I think you should include that aspect.
> > >
> > > Thank you for pointing this out. This value is **not** a typo. This drastic performance drop is due to the fact that, when $\alpha = 1$, the attention term $(1-\alpha) softmax(H)V$ is completely removed from the MHA calculation, turning the MHA module into a simple identity map (Figure 1(a)). This results in a significant loss of learning ability.
> > >
> > > **Regarding the difference in embedding structures between self-attention and MHA**
> > > In the CIFAR10/100 experiment, low-resolution 32x32 images were resized to 224x224 and input to ViT. This upsampling process makes information about neighboring patch tokens redundant, tending to produce high cosine similarity in the initial layer.
> > >
> > > To clarify this point, we calculated cosine similarity statistics for MHA($\alpha = 0.7$) using the high-resolution dataset ImageNet-1k:
> > >
> > > | layer | Mode | Median    | Mean |
> > > |-------|---------------|--------|--------------|
> > > | 1     | -0.07       | -0.04 | 0.07      |
> > > | 2     | -0.09        | 0.00 | 0.02     |
> > > | 3     | -0.09       | 0.00 | 0.02 |
> > > | 4     | -0.05  |0.05 | 0.06 |
> > > | 5     | 0.00 | 0.03 | 0.05 |
> > > | 6     | -0.05 | 0.02 | 0.04 |
> > > | 7     | -0.13 | 0.01 | 0.01 |
> > > | 8     | -0.07 | 0.04 | 0.05 |
> > > | 9     | -0.04 | 0.01 | 0.04 |
> > > | 10    | -0.14 | -0.03 | 0.01 |
> > > | 11    | -0.16 | 0.01 | 0.01 |
> > > | 12    | -0.13 | 0.06 | 0.04 |
> > >
> > > These results show that the cosine similarity of the first layer significantly decreases in ImageNet-1k. We also confirmed that the sharp peak around cosine similarity 1 seen in CIFAR has been eliminated. We plan to incorporate this in the camera-ready version, including more detailed visualization (e.g., violin plots).
> > >
> > > **Regarding the separation of dynamics in FFN and MHA, and the energy function**
> > > > Certainly, we can see the benefits of having separate dynamics (between FFN and MHA) here. Hence, please do discuss this more in-depth in your final version.
> > >
> > > Thank you for highlighting this important point. The absence of an energy function in our proposed MHA is theoretically important issue, and we hope to clarify this as a topic for future research. In the camera-ready version, we will expand on this discussion, including comparisons with existing related works.
> > >
> > > **Regarding the image quality of Figure 1**
> > > > I think Fig. 1 quality is bad.
> > >
> > > We apologize for the inconvenience. In the camera-ready version, we plan to avoid raster formats such as JPEG and replace them with high-resolution vector formats.
> > >
> > > Once again, we sincerely thank you for your continued engagement and constructive suggestions. Please feel free to reach out with any further questions or comments.

---

### Official Review · Reviewer_CUjJ · 2025-06-29

**Clarity:** 2
**Significance:** 2
**Originality:** 2
**Rating:** 3
**Confidence:** 3

**Summary:**

This paper introduces a novel perspective by interpreting the dynamics of hidden states in modern Hopfield networks as a form of transformer attention. The resulting mechanism, termed Modern Hopfield Attention (MHA), is shown—both theoretically and empirically—to mitigate issues such as rank collapse and token uniformity that affect standard self-attention. By replacing the self-attention modules in architectures like GPT-2 and Vision Transformer (ViT) with MHA, the authors demonstrate improved performance on text generation and image recognition tasks.

**Questions:**

1. The authors refer to the visible variables as $v$ in the text, but use $x$ in Equation (1). Are these variables intended to represent the same quantity? Clarifying the notation would help avoid confusion.

2. Why do the results in Tables 2 and 5 differ significantly, despite both using the same datasets (CIFAR-10 and CIFAR-100)? A more detailed explanation of the experimental settings and differences is needed. In general, the table captions should provide clearer and more comprehensive descriptions to aid interpretation.

**Ethical Concerns:**

["NO or VERY MINOR ethics concerns only"]

**Final Justification:**

After reviewing the responses and related discussions, I have decided to maintain my original rating of 3 (borderline reject), as I remain unconvinced that the new formulation has a substantial impact on practical performance.

**Limitations:**

yes

**Paper Formatting Concerns:**

The overall flow of the paper could be improved for better clarity and coherence. Currently, empirical results are presented early on, followed by theoretical explanations in Section 4 about how MHA improves transformers, and then additional empirical results. Reorganizing the content so that theoretical insights precede or are more tightly integrated with the experiments may enhance readability.

Additionally, Section 5 (Related Work) could be strengthened and repositioned earlier in the paper—preferably right after the Introduction—to more effectively contextualize the contribution within existing research and provide a coherent background for readers.

**Quality:**

2

**Strengths And Weaknesses:**

Strengths:
The paper introduces a novel perspective by utilizing the full dynamics of hidden states in modern Hopfield networks, moving beyond the commonly adopted adiabatic approximation. In addition, the authors provide solid theoretical analysis demonstrating that the proposed Modern Hopfield Attention (MHA) mechanism effectively mitigates rank collapse—a known limitation of standard self-attention.

Weaknesses:
Despite the theoretical contributions, the empirical improvements in text generation and image recognition tasks appear relatively modest compared to models using conventional self-attention. Moreover, the performance is highly sensitive to the hyperparameters $\alpha$ and $\alpha'$, as evidenced in Table 4. Given the degree of accuracy fluctuations, it becomes questionable whether the small performance differences reported in Tables 1, 2, and 3 are statistically meaningful.

---

> ### Author Rebuttal · Authors · 2025-07-31
>
> First of all, we would like to express our gratitude for your valuable comments and review. We also appreciate your evaluation of our theoretical contributions.
>
> **[Weakness]**
>
> As you pointed out, in the experiments in Table 4, we investigated $\alpha, \alpha’$ in the range of 0 to 1, and the behavior of the network changes dramatically in this parameter range. Specifically, $\alpha=1$ corresponds to a pure feed-through layer without any attention, $\alpha=0$ corresponds to a pure attention layer that removes shortcut paths, $\alpha’=1$ corresponds to a case where attention is calculated using only information from the previous layer, and $\alpha’=0$ corresponds to conventional self-attention.
>
> Despite these structurally different behaviors, the performance variation is relatively mild (about 2% in the range of $\alpha,\alpha’$ from 0.1 to 0.8), and the key point of this table is that sufficient performance can be obtained with the simple setting of $\alpha=\alpha’=0.5$. This demonstrates that our proposed method is robust and does not rely heavily on tuning (The effect of hyperparameter tuning is relatively mild).
>
> Furthermore, we present additional results that suggest the possibility of further performance improvements through technical innovations.
>
> First, regarding the Top-1 performance on ImageNet-1k, we tested the hyperparameter $\alpha=\alpha'=0.7$ on ImageNet-1k, similar to the CIFAR experiment (Table 2). As a result, the performance difference became even more pronounced:
>
> Additional results on ImageNet-1k (Top-1 Accuracy)
> | Self-attention | MHA($\alpha=0.5$) | MHA($\alpha=0.7$) |
> | ---- | ---- |---- |
> | 76.074 | 76.434 | 77.058 |
>
> The performance improvement of over 1% without adding any training parameters is sufficiently meaningful when compared to the improvement achieved by modifying ViT-B in existing studies [39][50] (both less than 0.6%). These existing studies differ from this paper in that they achieved accuracy by introducing additional learning parameters, making this performance difference even more notable.
>
> Additionally, as an additional experiment, we report the results of applying MHA to the Llama architecture (perplexity). In the following experiments, we trained Llama on Wikitext103 as well as various datasets with distinct characteristics, such as books and news articles.
>
> Additional results in Llama (Perplexity)
> | Dataset |  Self-attention | MHA(\alpha=0.5) |
> | ---- | ---- | ---- |
> | Wikitext103(20%)| 29.92| **24.48** |
> | BookCorpus(20%)| 67.05 | **40.68** |
> | CNN_dailymail(20%)| 78.81| **58.62** |
> | xsum(20%)| 103.30| **97.91** |
>
> The above results were obtained by comparing different language domains using the mini-llama implementation. In all cases, MHA outperformed the baseline, demonstrating that our method is effective in terms of both architecture and domain.
>
> **[Question 1]**
>
> Thank you for pointing that out. As you mentioned, $v$ is a typo for $x$ and means the same variable. We will correct it in the camera-ready version.
>
> **[Question 2]**
>
> Thank you for your important question. Table 2 shows the results of applying MHA to a conventional ViT model, while Table 5 shows a comparison using an attention-only model that excludes shortcut paths and FFNs in order to approximate our theoretical setup.
> As reported in previous studies [10], rank collapse is particularly prominent in attention-only models, so we adopted this setting to more clearly demonstrate the effectiveness of MHA. This point is explained in lines 270–279, but to make it clearer, we will revise the caption of Table 5 to specify the experimental conditions.
>
> **[Comments on Formatting]**
>
> Thank you very much for your suggestions on the structure. As you pointed out, we will consider summarizing the theoretical core results (e.g., Theorem 2) in the introduction to aid the reader's understanding.
> In addition, we will rearrange the related research (Section 5) immediately after the introduction and reconsider the structure to more clearly convey how our contributions relate to existing literature.

---

> > ### Comment · Reviewer_CUjJ · 2025-08-05
> >
> > Thank you for your responses. I appreciate your clarification regarding the variation of $\alpha$ and $\alpha'$. You mentioned that this variation does not lead to performance differences, highlighting the robustness of the method. However, from another perspective, this robustness implies that the variation is largely insignificant. Therefore, while I acknowledge the theoretical contribution of your modern Hopfield attention mechanism, it appears to have a limited impact on performance.
> >
> > My second question concerns your new experiment on Llama. I wonder why the BookCorpus and CNN_dailymail datasets show a greater improvement, in contrast to the more modest gains observed with Wikitext-103 and xsum.

---

> > > ### Author Response · Authors · 2025-08-07
> > >
> > > Thank you again for your thoughtful comments.
> > >
> > > **robustness of the method**
> > >
> > > As acknowledged by the reviewers, the primary goal of this paper is not to propose a new practical method, but rather to deepen the theoretical understanding of the role of the hidden state in Hopfield networks. On that basis, the experiments on multiple benchmarks are intended to confirm that the proposed MHA structure can form a meaningful variant of the Transformer, rather than to maximize performance itself.
> > >
> > > We understand your concern that the observed robustness of our method for $\alpha$ and $\alpha'$ might suggest the limited practical effect of these variables. However, conversely, this point also proves that our method does not rely on fragile hyperparameter tuning and has robust design principles. We believe that such characteristics are of great value in large-scale architecture design.
> > >
> > > Furthermore, the purpose of the CIFAR10/ViT-T experiment was to simply understand that a certain level of performance can be achieved even with a very simple hyperparameter setting ($\alpha = \alpha' = 0.5$). Therefore, we do not intend to use these results to discuss the general performance improvement of MHA.
> > >
> > > At the same time, the fact that the proposed method was effective even on well-designed architectures used in real-world applications, such as Llama, suggests that additional performance improvements can be achieved even on existing complex architectures by simply incorporating MHA. We believe it has potential for future applications.
> > >
> > > **on new experiment**
> > >
> > > Regarding your question about why significant improvements were observed on some datasets, in this Llama experiment, our primary goal was to confirm the positive impact of the proposed method within a limited timeframe. Therefore, we did not perform hyperparameter tuning for each dataset (e.g., learning rate or number of training steps). As a result, the performance varies considerably depending on the difficulty and characteristics of each dataset.
> > >
> > > As we have informed the other reviewers, we plan to include results from the entire dataset in a camera-ready version.
> > >
> > > Finally, although the main objective of this work is to provide a theoretical understanding of the role of hidden states, we firmly believe that this foundational insight will contribute to future performance improvements. We sincerely appreciate your continued consideration.

---

> > > > ### Comment · Reviewer_CUjJ · 2025-08-09
> > > >
> > > > Thanks for the response. I have no further questions.

---

> > > > > ### Author Response · Authors · 2025-08-09
> > > > >
> > > > > We sincerely thank the reviewers for their thoughtful comments and suggestions, and we greatly appreciate your time and constructive cooperation throughout this peer review process.

---

### Official Review · Reviewer_WaSH · 2025-07-02

**Clarity:** 3
**Significance:** 2
**Originality:** 3
**Rating:** 4
**Confidence:** 3

**Summary:**

This paper proposes a new continuous modern Hopfield network architecture that introduces dynamics in the hidden state. The number of parameters is kept constant but the computational time is increased linearly as a function of the transformer dimension. The authors show this architecture prevents the rank collapse problem, which may be what modestly improves the ViT and GPT performance.

**Questions:**

Can the authors identify stronger improvements over baselines using their proposed principle? The improvements seem relatively modest compared to the sophistication of the theory but happy to be convinced otherwise.

I think the figure quality could be significantly improved.

**Ethical Concerns:**

["NO or VERY MINOR ethics concerns only"]

**Final Justification:**

I think the author's rebuttal points are fair in how they contextualize their contribution. As I mention - I am sympathetic to their theoretic perspective (and I really like this paper conceptually) however the empirical results suggest modest benefits (that, ultimately, may not be statistically meaningful) and Im not sure that the authors' have demonstrated the core contribution they propose in terms of performance. Taken together, I think that my rating is appropriate.

**Limitations:**

yes

**Paper Formatting Concerns:**

The figures seem of relatively low quality/resolution especially Figure 2.

**Quality:**

3

**Strengths And Weaknesses:**

The derivation of the hidden state dynamics seems sound.The authors clearly describe previous and related work. X and Y labels in Figures 1(b) and 2 are missing. Would be important to have confidence intervals in the perplexity results throughout the paper, to check if the improvements in perplexity are statistically significant.

The paper is clear and well organized.

Addressing the rank collapse problem is important and the theoretical link to MHN is intriguing via the consideration of the dynamics in the hidden state allows for an improvement of the rank collapse problem. However, it would be nice to actually show that the rank is not collapsing in addition to showing the activation of selected layers is more uniform compared to the self attention case. Ultimately, it seems that the performance improvements are modest.

Is it clear how this work differs from previous contributions. The proposed algorithm improves slightly the performance compares to the previous state of the art. The paper provides new insights on how to use modern hopfield networks in transformer architectures.

---

> ### Author Rebuttal · Authors · 2025-07-31
>
> We would like to express our sincere gratitude for your deep understanding of this paper and your numerous constructive and useful comments.
>
> **[Weakness] Confidence intervals for perplexity and figure labels**
>
> At present, due to the computational cost of a single training run, we have not obtained sufficient statistics. By the camera-ready version, we will obtain statistical metrics (mean and standard deviation) based on multiple runs and clearly indicate their significance.
>
> On the other hand, as an alternative to confidence intervals, we conducted the following additional experiments to demonstrate that perplexity improvements in language models can be systematically reproduced in other models and datasets. Using the mini-llama implementation, we evaluated the perplexity of Wikitext103, BookCorpus, CNN/DailyMail, and XSum (20% each), and found significant improvements in all cases.
>
> Additional results in Llama (Perplexity)
> | Dataset |  Self-attention | MHA(\alpha=0.5) |
> | ---- | ---- | ---- |
> | Wikitext103(20%)| 29.92| **24.48** |
> | BookCorpus(20%)| 67.05 | **40.68** |
> | CNN_dailymail(20%)| 78.81| **58.62** |
> | xsum(20%)| 103.30| **97.91** |
>
> These provide strong evidence of reproducible improvements across architectural domains.
>
> We also appreciate your comments regarding the lack of axis labels in Figures 1 and 2, as well as the overall quality of the figures. In the camera-ready version, we will replace the figures with high-resolution versions that clearly show the axis labels and legends.
>
> **[Weakness] Rank collapse and changes in activity**
>
> Rank collapse refers to a decrease in the rank of the representation matrix and is closely related to token uniformity (all tokens becoming similar vectors), but they are not strictly the same. For example, even if the rank is small and the vectors are aligned, if there is variation in the direction between the vectors, it cannot be called uniformity. In this study, to avoid confusion between the two, we use cosine similarity distribution to investigate token uniformity.
>
> To address your question, we will report not only the angles (similarity) between tokens but also the statistics of their norms. Since norms represent a different degree of freedom from angles, there are various possibilities, such as the cosine approaching 1 and aligning while the norms remain highly variable.
>
> The following are the average values (mean) and variances (std) of token norms for each layer of trained ViT-L and MHA. Interestingly, while MHA resolved token uniformity, it was observed that it also uniformly aligned the norms of each token and greatly reduced the variance. Thanks to your comments, we discovered a very interesting phenomenon, which we would like to pursue as an interesting research topic in the future.
>
> Cifar100, ViT-Large
> | layer | Mean(Self-attention) | Std(Self-attention)        | Mean(MHA,$ \alpha=05$) | Std(MHA, $\alpha=05$) | Mean(MHA, $\alpha=07$) | Std (MHA,$\alpha=07$) |
> |-------|---------------|-----------|--------------|-----------|--------------|-----------|
> | 1    | 10.4820      | 0.5994   | 31.9970     | 0.0003455 | 31.9970     | 0.0003455 |
> | 4    | 13.6818      | 0.2291   | 31.9970     | 0.0000852 | 31.9970     | 0.0000852 |
> | 8    | 13.9900      | 0.1034   | 31.9970     | 0.0000563 | 31.9970     | 0.0000563 |
> | 12   | 13.5533      | 0.0630   | 31.9969     | 0.0000698 | 31.9969     | 0.0000698 |
> | 16   | 13.4659      | 0.1685   | 31.9968     | 0.0000487 | 31.9968     | 0.0000487 |
> | 20   | 13.4335      | 0.4192   | 31.9968     | 0.0000751 | 31.9968     | 0.0000751 |
> | 24   | 11.4047      | 0.9176   | 31.9967     | 0.0000690 | 31.9967     | 0.0000690 |
>
> Similarly, when we evaluated GPT-2, we found no significant differences between GPT and MHA. This is likely due to the fact that the detailed behavior of MHA differs between the bidirectional and autoregressive Transformers.
>
> Wikitext103, GPT-2(medium)
> | layer | Mean(Self-attention) | Std(Self-attention) | Mean(MHA, $\alpha=05$) | Std(MHA, $\alpha=05$) |
> |-------|---------------|--------|--------------|--------|
> | 1     | 1.3457        | 0.6444 | 1.4607       | 0.4460 |
> | 4     | 10.6074       | 3.0606 | 7.5750       | 1.9212 |
> | 8     | 2.6345        | 0.6598 | 0.2133       | 0.0575 |
> | 12    | 3.1324        | 0.6150 | 3.0704       | 0.6068 |
> | 16    | 13.1777       | 8.8199 | 20.2960      | 8.1086 |
> | 20    | 18.3072       | 6.0310 | 17.8403      | 4.0680 |
> | 24    | 19.1561       | 2.9285 | 17.1602      | 1.8057 |
>
> **[Weakness, Question] The usefulness of performance improvement through MHA**
>
> I understand your question to be about the significance of the proposed algorithm among the numerous technical improvements made to Transformers to date.
>
> As you understand, the main focus of this paper is not to propose a new practical method, but to understand the role of hidden states in Hopfield Networks from the context of Transformers. To demonstrate that the attention mechanism (MHA) derived from the discussion in this paper has a meaningful structure, we confirmed performance improvements in language and image tasks. These results also suggest practical usefulness, so we agree that the significance of comparing it to existing architectures, as in your question, is an important point.
>
> Regarding your concern about the “relatively modest" performance improvement, there is potential for further improvement through hyperparameter tuning and technical refinements. As evidence of this, we present the results of additional experiments. Similar to CIFAR experiments, using the setting $\alpha=\alpha’=0.7$ on ImageNet-1k further clarified the performance improvement:
>
> | Self-attention | MHA($\alpha=0.5$) | MHA($\alpha=0.7$) |
> | ---- | ---- |---- |
> | 76.074 | 76.434 | **77.058** |
>
> In this way, it is suggested that further performance improvements are possible through appropriate hyperparameter selection.
>
> Another important result of the additional experiments is the systematic and significant performance improvement over Llama shown above. The fact that applying MHA to an architecture like Llama, which has been technically optimized, still improves performance suggests that simply adding MHA to the various Transformer variants proposed so far for performance optimization could achieve further performance improvements and contribute to raising the state-of-the-art. While a large-scale verification of MHA is beyond the scope of this paper, it is considered an extremely interesting research direction.
>
> **[Quality of Figures]**
>
> We apologize for any inconvenience caused by the insufficient resolution of the figures. We will create high-quality figures and replace them in the camera-ready version.

---

> ### Comment · Reviewer_WaSH · 2025-08-04
> **follow-up**
>
> Thanks for the clarifications - I do not have further questions.
>
> Regarding the practical usefulness issue - yes, you understand my query correctly, apologies for the brevity of my comment on this. I am sympathetic to this view i.e. focusing on a theoretic question of hidden states rather than practical performance per se. Potentially, it would be useful to elaborate the points you make in this regard as limitations/future directions.

---

> > ### Author Response · Authors · 2025-08-06
> >
> > Thank you very much for your understanding and thoughtful comments. I sincerely appreciate your engagement with this research from a theoretical perspective. As you rightly pointed out, one of the central motivations of this work is not only to improve practical performance, but also to explore the theoretical role of hidden states in Transformers more deeply.
> >
> > As you suggested, we will make sure to clearly express this point of view as part of our discussions about the limitations and future direction in the camera-ready version. Once again, thank you very much for your valuable feedback and support.

---

### Official Review · Reviewer_aW1E · 2025-07-02

**Clarity:** 2
**Significance:** 3
**Originality:** 4
**Rating:** 5
**Confidence:** 4

**Summary:**

The paper relooks at the connection between Modern Continuous Hopfield Networks and the Attention Mechanism, as in some relevant prior work (mentioned in the paper). A traditional Hopfield Network consists of two layers, the feature neurons and the hidden neurons (denoted by $v$, $h$ respectively). In that context, it identifies a common assumption made in most contemporary work of discretizing the evolution of the dynamic system under the adiabatic limit and chooses NOT to assume this. Not making this assumption leads to an interesting modified attention style update rule, that performs the softmax operation on the feature neurons, which is in turn updated via a recurrence equation involving its own previous state. Thus a skip connection is implicitly introduced in the proposed attention mechanism. This new rule is rigorously derived from the dynamics of MCHN and tested on a few standard models cum datasets, in both vision and language. Finally, the impactful utility of this attention scheme is demonstrated in the context of token uniformity/rank collapse, a common problem in vanilla Transformers, both theoretically and empirically.

**Questions:**

1. Are rank collapse and token uniformity different names for the same problem? Or are they different problems that always occur together?

**Ethical Concerns:**

["NO or VERY MINOR ethics concerns only"]

**Final Justification:**

The paper presents an interesting modification to self-attention based on a commonly ignored observation in energy based memory models, shows proof-of-concept level experiments and moderate theory to justify the contribution. While the work is probably still not full ripe in this form, I think it has good potential in general. The authors also did a great job clearing practically all my concerns. Thus, I hope to see this paper accepted at NeurIPS.

**Limitations:**

Yes.

**Paper Formatting Concerns:**

None.

**Quality:**

3

**Strengths And Weaknesses:**

Strengths:
1. The insight of NOT taking the adiabatic limit of the energy dynamics, then deriving an update rule based on the chosen Lagrangians, and then formulating a new attention mechanism is novel, and particularly so in the context of its connection to Hopfield Networks.
2a. This formulation involving hidden state recurrence is rigorously derived from the energy dynamics, and not just a variant of mechanisms that combine RNNs with Attention via some heuristics.
2b. Appendix D provides a case by case analysis for different choices of the discretization step which seems novel, interesting and potentially useful for future work as well.
3. The utility of this problem in handling a particular issue in training vanilla Transformers known as rank collapse is identified via some experiments.
4. A theoretical result is also suggested (Theorem 4.2) which suggests how the proposed attention mechanism, MHA, beats vanilla attention at handling rank collapse.
5. Overall, the presentation makes the paper easy to follow.

Weaknesses:

Experiments:
The experiments are overall quite weak to build confidence in the efficacy of the proposed methodology.

1. Especially for language, testing on one dataset alone is insufficient. I strongly suggest the authors to try on atleast 3 more moderate-sized language datasets commonly used in contemporary research such as OpenWebText, MMLU, GLUE, BookCorpus, etc. Based on other statements in the paper, I understand that compute is concern, but please try to choose datasets wisely such that it is not too compute-hungry while at the same time, the point is properly conveyed.

2. Same goes with the language models, currently only GPT-2 small and medium have been experimented. Please experiment with at least 2 more language models, e.g. amongst Llama, Gemma, GPT-3, Mistral, RoBERTa, or anything else convenient but authentic.

3. For images too, it is a similar story. Though three datasets have been experimented with, the improvement in case of CIFAR-10 and ImageNet-10K appears questionable. Further the claim made in lines 185-187 without a proper reference is hard to believe, and it better be backed by some reference or some experiments if that is true. Especially given that the results are not upto the mark for CIFAR-10 and ImageNet-10K, I suggest trying on at least two more datasets, such as TinyImageNet or any other relevant, well-known dataset and showing strong results on them.

4. I do not see any ablations/discussion on the initializations used in training the Transformers, for language and vision alike. This may be important in assessing the efficacy of MHA, as it may affect the results either way. Please try different initializations, including pretrained initializations for both vision and language (for instance, for ViTs, try LION initialization, CLIP based, Xavier-Glorot, etc.)

Technical:

1. Lines 246 and 254-256 suggest that a doubly-exponential decay is avoided because the case $m=0$ appears in one of the arguments to the max function. However I don't think that directly holds, because it is very well possible that the other terms with $m \neq 0$ dominate the overall max function and thus becomes doubly-exponential.

2a. Even if $||Res(AttnNet(X))||$ decreases linearly as Theorem 4.2 suggests, it is not necessary that it happens because of MHA. Because ultimately, both Theorems 4.1 (which is not the authors contribution, as far as I understand) and Theorem 4.2 (which is a contribution from the authors, if I understand correctly) provide upper bounds, so it seems possible that even without MHA, $||Res(AttnNet(X))||$ was lower than the (potentially stronger) upper bound that Theorem 4.2 provides, which would make the theorem redundant in that case.

2b. Ideally, to show the efficacy of MHA at avoiding rank collapse, there should be at least one example shown where under standard training of vanilla Transformers with vanilla attention, $||Res(AttnNet(X))||$ is lower bounded by some value, say $\kappa$, and by using MHA for that case, $||Res(AttnNet(X))||$ can be brought down below $\kappa$. The more examples this phenomena holds for, the better the efficacy of MHA.

3. Equation 16 in the Appendix looks wrong -- in particular I don't see how the second term in the LHS appears, and also how the terms represented by '...' are constant terms. Please fix this, as this seems to be a crucial equation for the rest of the proof to go through.

Presentation:
1. Please rigorously define the exact expression for token similarity, as described in lines 280-282. Please also more clearly explain what exactly the violin plots are plotting, as it not is very clear in the current writeup. Also mention how many tokens are used in describing a single such plot.

2. Please move lines 181-183 of the Appendix or some equivalent of it into the main text, as this more clearly describes what token uniformity is, particularly in the context of rank collapse. I don't see that clear explanation given elsewhere in the main text.

Minor:

1. In line 14 of the abstract, please change "problem" to "problem$\textbf{s}$".
2. In line 40 of the main paper, please change "lack" to "lack$\textbf{s}$".
3. Please replace the 3 places in the main text where "chapter" is used with "section"/"subsection" or some more appropriate word like that.
4. Please use better notation for the max function in the statement for Theorem 4.2.

P.S: I look forward to increasing my score, especially if concerns stated in the Experiments section, first two concerns in the Technical & Presentation sections are addressed. I would thus advise the authors to prioritize addressing the concerns in that order.

---

> ### Author Rebuttal · Authors · 2025-07-31
>
> We sincerely appreciate your valuable review and helpful comments. We are particularly grateful for your positive evaluation of the novelty and theoretical contributions of this study.
>
> **[Experiment: W1, W2]**
>
> As you are probably already well aware, the main purpose of this paper is not to propose a new technical method, but to understand the role of hidden states in Hopfield Networks from the perspective of Transformers. In that context, we conducted empirical verification through performance changes in language and image tasks and mitigation of rank collapse to demonstrate that the corresponding attention mechanism (MHA) has a meaningful structure.
>
> However, we fully agree that the reliability and comprehensiveness of the experiments are important issues. Therefore, in response to your comments, we conducted additional experiments within the constraints of computational resources, including a more realistic architecture (Llama) and language datasets from domains other than Wikitext103. The data used were selected from standard datasets available on Hugging Face Datasets, such as BookCorpus, cnn_dailymail, and xsum. These datasets, which have characteristics different from Wikitext103, such as books and news articles, are suitable for verifying language domain generalization. The following table shows the comparison results of the validation perplexity of Llama trained on each dataset. We used the mini-llama codebase as a reliable and convenient implementation. In the following experiments, unless otherwise specified, MHA is set to $\alpha=0.5$ for all cases.
>
> **Wikitext103(20%) step=40k**
> | Self-attention(Llama)  | MHA(Llama)  |
> | ---- | ---- |
> | 29.92| **24.48** |
>
> **BookCorpus(20%) step=6k**
> | Self-attention(Llama)  | MHA(Llama) |
> | ---- | ---- |
> | 67.05 | **40.68** |
>
> **CNN_dailymail(20%) step=6k**
> | Self-attention(Llama)  | MHA(Llama) |
> | ---- | ---- |
> | 78.81| **58.62** |
>
> **xsum(20%) step=20k**
> | Self-attention(Llama)  | MHA(Llama)  |
> | ---- | ---- |
> | 103.30| **97.91** |
>
> **Training dataset included with the mini-llama implementation**
> | Self-attention(Llama)  | MHA(Llama)  |
> | ---- | ---- |
> | 1.292| **1.28152** |
>
> These results demonstrate that MHA is effective not only for simple GPT series but also for the latest architectures such as Llama, and that systematic performance improvements can be observed across domains. Thus, we believe that our claims are not limited to results specific to a single model or dataset, but have broader validity.
>
> **[Experiment: W3]**
>
> We apologize for failing to adequately convey the significance of the results due to the lack of references supplementing our understanding that a 1% improvement in accuracy on ImageNet-1k is non-trivial.
>
> First, similar to our research, methods that correct attention maps, such as DeepViT [50], have reported performance improvements of approximately 0.3% for models equivalent to ViT-B ([50], Figure 1, 12-layer). Additionally, the improvement margin for ViT-B in [39] is +0.6%. While both of these methods involve adding learning parameters, our MHA achieves the approximately 1% improvement shown below without adding any parameters. We consider this to be a technically significant result.
>
> Furthermore, improvements of over a few percent on ImageNet typically require distillation, large-scale pre-training, or significant design improvements. For example, the +4% improvement from ViT-B/16 to DeiT-B is due to the distillation effect described in [Touvron et al., ICML 2021]. Such results were obtained through design explicitly aimed at performance improvement, and cannot be directly compared with our results, which are based on minor modifications driven by theoretical motivations.
>
> Furthermore, to confirm the robustness of the performance improvement, we conducted additional ViT-B on ImageNet experiments with the same setting of $\alpha=\alpha'=0.7$ on ImageNet-1k as on CIFAR:
>
> | Self-attention | MHA($\alpha=0.5$) | MHA($\alpha=0.7$) |
> | ---- | ---- |---- |
> | 76.074 | 76.434 | **77.058** |
>
> As shown above, the effect of performance improvement becomes even clearer with the appropriate selection of hyperparameters.
>
> **[Experiment: W4]**
>
> Thank you very much for your keen observation regarding initialization. In this study, we used Pytorch's nn.Linear to calculate Q, K, and V, and adopted Kaiming-Uniform for initialization. On top of that, we conducted additional comparative experiments with Xavier-Glorot initialization.
>
> **Comparison of initialization: Cifar100, MHA(ViT-tiny), epoch300**
> | Xaiver-Glorot | Kaming-He|
> | ---- | ---- |
> | 68.34| 64.00 |
>
> The improvement in performance achieved through Xavier initialization is intriguing, and there is potential for further performance gains through initialization. Such technical innovations represent an exciting direction for future research.
>
> Furthermore, initialization using pre-trained weights involves applying weights trained with different model (plain self-attention) to a new MHA, which may affect learning stability. Therefore, the development of various training techniques is necessary (e.g., $\alpha$ warm-up). In this study, we focused on structural understanding based on theoretical motivation, so we refrained from conducting such experiments this time, but we consider this to be an important direction for future research.
>
> **[Technical: W1]**
>
> All terms other than $m=0$ in lines 246, 254-256 are also positive. Therefore, the right side of our inequality evaluation obtained by adding these terms is at least greater than the term $m=0$. Thus, even if the other terms decay exponentially, the term $m=0$ dominates, preventing exponential decay on the right side as a whole. Regarding this point, we will add a clear explanation in the camera-ready version.
>
> **[Technical: W2a, 2b]**
>
> The theoretical contribution of this study is based on an “upper bound evaluation of rank collapse” in the context of previous studies ([10], [39], etc.). It is extremely difficult to provide a mathematical lower bound, and such an evaluation remains an unsolved problem at this point. Therefore, this study endeavored to complement the theoretical results with empirical evidence.
>
> Furthermore, previous studies have sought architectural innovations to improve the upper bound of the inequality. This approach suggests architectural structures that may mitigate double exponential decay based on theoretical insights (e.g., the effects of shortcut paths and MLP in [10], or modifications to shortcut paths in [39]). This paper also follows this logic.
>
> For example, Table 5 shows that rank collapse is mitigated by MHA in the same attention-only network as the theoretical setup. Furthermore, the cosine similarity distribution in Figure 2 shows a similar trend, providing evidence that bridges the gap between theory and actual networks.
>
> Furthermore, stable learning of the attention-only network itself is difficult (Table 5), and it is impossible to make a meaningful comparison of $\|Res(AttnNet(x))\|$ with an untrained deep model. Therefore, we supplement the validity with observations of token uniformity improvement not only in the attention-only net but also in the actual Transformer.
>
> **[Technical: W3]**
>
> There is no error in Appendix (16). The second term you pointed out is the result of applying (18) to $1 b_{Q}^{(\ell) T} W_{K}^{(\ell) T} X^{(\ell) T}$. Furthermore, the last item in Appendix (16) can be written explicitly as
>
> $$
> \frac{1}{\sqrt{d_k}}( X^{(\ell)} W_Q^{(\ell)} b_K^{(\ell)} + 1 b_Q^{(\ell) T} b_K^{(\ell) })1^T
> $$
>
> The reason why this last term can be ignored is that the row-wise softmax function is invariant to constant shifts in the logits, as shown by $e^{A_j+c}/\sum_i e^{A_i+c}=e^{A_j}/\sum_i e^{A_i}$ (see lines 48-50). In fact, the term described above takes the form of the outer-product of two vectors, $v 1^T$, and thus serves only to add a constant to each row of $A$.
>
> **[Presentation W1]**
>
> The cosine similarity was calculated using the following procedure:
>
> 1. Extract all tokens $x_n(n=1,…,N)$ in a specific layer.
> 2. Calculate the cosine between the two vectors for all pairs $(x_n, x_m)$.
> 3. Generate a violin plot from these cosine values collected from multiple samples.
>
> Number of tokens used:
>
> - ViT: 256 tokens $\times$ 32 samples = 8192 tokens
> -  GPT-2: 1024 tokens $\times$ 12 samples = 12288 tokens
>
> **[Presentation: W2, Q]**
>
> As you pointed out, we will move the explanation of token uniformity to the main text.
> Rank collapse refers to “a phenomenon in which the rank of the representation matrix decreases significantly,” and narrowly defined token uniformity (all tokens becoming similar vectors) is one form of this. However, even if the rank is small, if there is variation in the direction between vectors (e.g., opposite directions), it cannot be called uniformity. In this study, to avoid confusion between the two, we explicitly observe uniformity using the cosine similarity distribution.
>
> **[Minor Issues]**
>
> All the description and notation errors you pointed out will be corrected in the camera-ready version.
>
> **Thank you very much for your valuable comments.**

---

> > ### Comment · Reviewer_aW1E · 2025-08-04
> > **Encouraging Responses, Needs Little More Work**
> >
> > I thank the authors for their efforts to answer all my concerns, in a short amount of time. I apologize for responding a little late. Most of the responses are satisfactory and better bring out the contributions of this paper. Below I will only list the points that I feel still need more explanation (in descending order of importance):
> >
> > 1. [Technical, W1] I think it is fair to believe that the exponential terms will be the resultant term after applying $\verb|max|$ over all those arguments from $m=0 \ldots L$. I don't quite follow how the $m=0$ term will dominate then.
> >
> > 2. [Technical, W2a,2b] I very much understand that theoretically deriving a "generic" lower bound of this nature could be very hard, even from personal experience. Therefore what I was suggesting was to take a very, very simple toy example e.g. a one-layered Transformer block, with all zeros initialization, and very simple ResNet architecture and see using vanilla GD (not SGD, since it may only become harder to prove with that) if you can show for that simple example that MHA as you propose is better suited to tackling the rank collapse problem, given the same number of epochs say. If this is also difficult, it is understandable, but please do mention this gap between what is desired and what the current bound conveys (1-2 lines about this is enough), with the citations that you posted on how all existing works suffer from this gap.
> >
> > 3. [Presentation W1] If the point is to show that similar vectors are more frequent, might that not be presented better via histograms rather than violin plots, where the frequency of vectors with a set similarity score are plotted?
> >
> > 4. [Experiment, W3] I agree with the points raised by the authors regarding this experiment and would urge them to include this explanation in their manuscript, if their paper get accepted at this conference. I nonetheless would still insist on trying to show clearly better results on at least **one** more image dataset (of the authors choice, but basically where the point is conveyed without having to explain too much regarding incremental performances).
> >
> > I look forward to substantially increasing my score if these points are amply addressed.

---

> > > ### Author Response · Authors · 2025-08-07
> > >
> > > We would like to once again thank the reviewers for their detailed and constructive comments and suggestions on our work.
> > >
> > > **1 [Technical, W1]**
> > >
> > > We also apologize for our insufficient explanation.
> > >
> > > In the equation you mentioned, the $m=0$ term takes the form
> > >
> > > $(r\alpha' C_2)^{L} \| Res(X) \|_{1,\infty}$,
> > >
> > > and the other $m \neq 0$ terms take the form
> > >
> > > $(r(1-\alpha')C_1)^{(3^m-1)/2} (r\alpha'C_2)^{3^m(L-m)} (\|Res(X)\|_{1,\infty})^{3^m}$.
> > >
> > >  The important thing to note here is that the $m \neq 0$ terms have a larger coefficient in the exponent because $3^m(L - m) > L$ (for $m = 1, 2, \cdots, L-1$), and therefore experience stronger damping. For this reason, it is generally thought that the max function is likely to select the term with the smallest damping, $m=0$.
> > >
> > > Of course, in special cases where $\|Res(X)\|_{1,\infty}$ is a special value, for instance, it cannot be denied that the $m \neq 0$ term (e.g., $m=1$) may be larger than the $m=0$ term. However, this is an exceptional case, and in any case, the fact remains that terms other than higher-order double exponential decay  (e.g. $m=L$)  terms are selected by the max function. This discussion follows the same logic as [10].
> > >
> > > Thank you for your valuable comment, and we realize that our simplification of the explanation was unhelpful to readers. We plan to revise the final version to be more thorough and clear.
> > >
> > > **2 [Technical, W2a,2b]**
> > >
> > > Thank you very much for your insightful suggestion.
> > > As the impact of the double-exponential norm decay becomes particularly prominent in deeper models, providing a clearer and more direct setting to compare with our theoretical analysis. Therefore, we believe that conducting evaluations in multi-layer settings is appropriate.
> > >
> > > After careful consideration of your suggestion, we have decided that additional experiments using the following metrics at initialization based on [10],
> > >
> > > $\| Res(AttnNet(X)) \|_{1,\infty} / \| AttnNet(X) \|_{1,\infty}$
> > >
> > > which we believe offers an informative perspective on the evaluation you pointed out. The change of this normalized norm (metric) across layer is shown below:
> > >
> > > **CIFAR-10, Attention-Only Network (ViT-T setting)**
> > >
> > > | layer | Self-attention | MHA(\alpha=0.5) |
> > > | ----  | ---- | ---- |
> > > | 1| 0.407969 | 0.87574506 |
> > > | 2| 0.055576194| 0.8612994 |
> > > | 3|  0.0006726028| 0.8122593 |
> > > | 4|  1.707888e-06|  0.7501045 |
> > > | 5|  1.7146715e-06|  0.72701323 |
> > > | 6|  1.7665825e-06|  0.6732544 |
> > > | 7|  1.6488774e-06|  0.62368613 |
> > > | 8|  1.7134173e-06| 0.58449733  |
> > > | 9|  1.6747372e-06| 0.520544  |
> > > | 10|  1.7223047e-06|  0.46531567 |
> > > | 11|  1.7381981e-06|  0.44375575 |
> > > | 12|  1.7086259e-06| 0.38132438  |
> > >
> > > The left column shows the results with standard self-attention. As observed in previous studies, we confirm a sharp norm decay across layers, corresponding to rank collapse. In contrast, the right column shows that introducing MHA significantly mitigates this norm decay and suppresses the rank collapse through layers.
> > >
> > > While the inequality we derive in this work provides a theoretical upper bound on the norm decay, it closely aligns with the empirical behavior, and captures the rank-collapse suppression effect of MHA. We sincerely appreciate your thoughtful comments and discussion.
> > >
> > > **3 [Presentation W1]**
> > >
> > > To capture the shape of the distribution as well as changes across models and layers, we chose to use violin plots. Since in rank collapse the vector norms do not necessarily match, directly comparing vector values is challenging. Therefore, we use cosine similarity as the metric for similarity.
> > >
> > > If your concern is that the violin plots may smooth the distribution shape and thus be insufficient as a visualization, we agree that there are limitations. For readers’ convenience, we will include histograms alongside the violin plots in the camera-ready version.
> > >
> > > **4 [Experiment, W3]**
> > >
> > > We appreciate your understanding of the experiment's intent. If our paper is accepted, we will explain this point in the manuscript and clarify the positioning of the ImageNet-1k results.
> > >
> > > We are also currently preparing to select and train additional image datasets. Although it is unclear whether we will be able to complete this in time for the discussion period, we plan to reflect results from these additional image datasets in the camera-ready version. We will keep you updated on any progress.
> > >
> > > We would like to thank you again for the opportunity to deepen our discussion.

---

> > > > ### Comment · Reviewer_aW1E · 2025-08-07
> > > > **Recurring Questions on Points 1, 4**
> > > >
> > > > I thank the authors for continuing to engage in productive discussions.
> > > > I approve of points 2, 3 and encourage the authors to make the appropriate changes in the manuscript. For point 4, I would have liked to see some preliminary results before the discussion ends, but I can understand that the limited turnaround time may make it difficult to come up with meaningful results. Accordingly, I can still keep my overall assessment of this paper high but might have to reduce my confidence a bit.
> > > >
> > > > For point 1, I am sorry if I am missing something trivial here, but I keep getting the same question repeatedly on this observation: how is the max function supposed to pick the term with the least damping? Is the max over the damping or something else?
> > > >
> > > > These concerns are holding me from increasing my score and improving my assessment of the paper, so please treat this on a priority basis.

---

> > > > > ### Author Response · Authors · 2025-08-08
> > > > >
> > > > > We sincerely thank the reviewer for the encouraging overall feedback and for pointing out the aspects that could benefit from further clarification. Below we provide additional explanations for the remaining points, prioritizing Point 1 as requested.
> > > > >
> > > > > **Point 1**
> > > > >
> > > > > We would like to add some explanation.
> > > > > To clarify, the max function in this context compares the numerical values of terms corresponding to $m=0,1,\cdots, L$.
> > > > > Following [10][39], we assume $rC_{1,2}, \| Res(X)\| < 1$. In practice, this is a natural assumption because $r=4H/\sqrt{d_k}$, and the dimension $d_k$ is typically much larger than the number of heads $H$ in real architectures.
> > > > > Under these conditions, for $m\neq0$ we have:
> > > > >
> > > > > $ (r(1-\alpha’)C_1)^{(3^m-1)/2} (r\alpha’C_2)^{3^m(L-m)}(\|Res(X)\|)^{3^m} < (r\alpha’C_2)^L\|Res(X)\|$
> > > > >
> > > > > Therefore,
> > > > >
> > > > > $max \big( (r(1-\alpha’)C_1)^{(3^m-1)/2} (r\alpha’C_2)^{3^m(L-m)}(\|Res(X)\|)^{3^m} , (r\alpha’C_2)^L\|Res(X)\| \big)= (r\alpha’C_2)^L\|Res(X)\|$
> > > > >
> > > > > which corresponds to $m=0$, the term with the smallest damping factor.
> > > > > In other words, under standard parameter regimes, the max function will typically select the $m=0$ term.
> > > > >
> > > > > **Other Points**
> > > > >
> > > > > Regarding Point 4, unfortunately the short review–response turnaround did not allow us to conduct the new experiments in a sufficiently reliable manner. We run the complete set of experiments and report them in the camera-ready version to fully address this point.
> > > > >
> > > > > We appreciate your pointing this out, and we will make sure to include these results in the camera-ready version.

---

> ### Comment · Reviewer_aW1E · 2025-08-08
> **Satisfied**
>
> This clears my concerns regarding the theoretical assertion. Please make sure this is clearly explained like this in the camera-ready version. As I mentioned earlier, since the specific experiment I asked for is not ready yet, I will have to reduce my confidence, but will increase my rating to a 5. Hope to see this paper accepted at NeurIPS. Good luck!

---

> ### Author Response · Authors · 2025-08-08
>
> Thank you very much for your insightful guidance. In the camera-ready version, we will include a clearer and more detailed explanation of Theorem 4.2, incorporating the valuable suggestion we have had.
>
> I would also like to express my sincere gratitude for your understanding of our research, your encouragement, and the tremendous support you have provided in clarifying the arguments of this paper through constructive discussions.

---

### Official Review · Reviewer_7EBm · 2025-07-02

**Clarity:** 3
**Significance:** 4
**Originality:** 3
**Rating:** 5
**Confidence:** 3

**Summary:**

This paper introduces a new type of self-attention layer in transformers called Modern Hopfield attention (MHA). Previous work had also noted the relation between Hopfield networks and transformers and  used this relation to develop Hopfield layers in transformers where the Hopfield network served as the memory in the layer.  Earlier it was shown that the state update rule of Modern Hopfield networks in the adiabatic approximation corresponded to self-attention in transformers. This result generalizes beyond the adiabatic approximation to show that the relationship still holds. Specifically, they use MHA to allow the inheritance of attention scores across the layers. The computational cost increase is modest compared to the cost of the computation of full attention in layers.
The key result is in the update rule of Equation (10) where instead of the attention layer, a hidden state propagates across the layer accumulating the attention score in each layer as an exponential moving average. The softmax of the hidden state gives the attention score.  Two parameters a and (1-a) control the behavior of the MHA.

**Questions:**

Since there is a recurrence formulation being implied with the hidden states, it would be good to see a discussion of this approach over the newer state-space models that are offering much more computational advantage. In other words, while the theoretical result is interesting, is it still relevant in the age of state-space models?


It would also be good to see a discussion comparing to the Hopfield layers proposed earlier in reference[35].

**Ethical Concerns:**

["NO or VERY MINOR ethics concerns only"]

**Final Justification:**

I have already indicated acceptance of the paper due to the reasons above mentioned in my original review.

**Limitations:**

Since there is no computational advantage of using the MHA layer and at best a 1% increase in performance in many cases, does the result have any practical implication? This would be good to point out in the discussion.
The related work section is bare-boned. An extensive discussion particularly w.r.t to other Hopfield layer formulations would be good to add to the paper. The paper itself did not point to any limitations.

**Paper Formatting Concerns:**

I didn't see any format violations

**Quality:**

4

**Strengths And Weaknesses:**

The derivations and the observations made between Modern Hopfield network and transformers are interesting. The paper is largely theoretical in nature and includes a computational complexity analysis that shows that there is a modest cost increase which is a function of the input sequence over and above the attention computation cost. The main novelty is in the modification of the attention layer as specified in Equation 10.

The contribution is an elegant theoretical contribution in which the attention weights are accumulated through the hidden Hopfield states in a sense combining recurrent neural networks within the architecture of transformers.  The impact of MHA on transformer performance was analyzed in downstream use cases of text generation and image recognition and in both cases, improvements of over 1% were seen, particularly for larger categories recognition.

---

> ### Author Rebuttal · Authors · 2025-07-31
>
> We sincerely appreciate your deep understanding of this research, your evaluation of its significance, and your insightful comments from a broad perspective.
>
> **[Question] Relationship with modern architectures (such as state-space models)**
>
> As you pointed out, the current trend in Transformer research is shifting toward linear structures such as linear memory models and state-space models (SSM). Even within this trend, we would like to explain the significance of this research.
>
> First, although not detailed in this paper, the proposed MHA method is also applicable to kernelized attention (linear attention). Linear attention is a framework that integrates RNN-like sequential computation with Transformer parallelism, and by adding gate mechanisms or masks, it can evolve into SSMs such as Mamba, RWKV, and RetNet. By integrating the state accumulation mechanism of MHA into linear attention, we believe it is possible to design SSMs that are both more performant and theoretically consistent.
>
> Furthermore, mapping the associative mechanism of the Hopfield Network to the Transformer corresponds to the repeated application of weight-shared attention layers. This is deeply related to structures such as the Universal Transformer (Dehghani et al., ICLR 2019) and the recent Looped Transformer (Yang et al., ICLR 2024). The “depth-wise recurrence” introduced in this study through MHA has an essential connection with such prior research.
>
> Going forward, we believe that hybrid models integrating layer recurrence via MHA and sequence-wise recurrence in SSM represent a promising direction for next-generation language models. Thank you for your valuable feedback.
>
> **[Question] Comparison with Ramsauer et al. [35]**
>
> I would like to add some additional information regarding the comparison with Ramsauer et al. [35]. In [35], the authors demonstrate the fast convergence of the Modern Hopfield Network (MHN) and justify its use as a non-recurrent layer in neural networks. On the other hand, in this study's MHA, we propose a dynamic structure that maintains and updates hidden states across layers.
>
> More specifically, the HopfieldPoolLayer and HopfieldLayer in [35] are formally similar to self-attention and do not have recursive properties, as they use the MHN output as a single output. In contrast, our MHA naturally incorporates Hopfield recursion into the Transformer layer structure, as “state accumulation and updating” are performed in each layer.
>
> We will clearly describe this difference in the camera-ready version as well.
>
> **[Limitations] Practical Significance and Applicability**
>
> As you pointed out, it is important to ask how much performance improvement the MHA proposed in this study will bring in practical terms. To clarify this point, additional experimental results in natural language tasks are shown below.
>
> **Additional results in Llama (Perplexity)**
> | Dataset |  Self-attention | MHA(\alpha=0.5) |
> | ---- | ---- | ---- |
> | Wikitext103(20%)| 29.92| **24.48** |
> | BookCorpus(20%)| 67.05 | **40.68** |
> | CNN_dailymail(20%)| 78.81| **58.62** |
> | xsum(20%)| 103.30| **97.91** |
>
> These results show that MHA brings about significant improvements in perplexity not only for standard GPT-2 but also for practical Llama-based architectures. Moreover, these improvements were achieved without additional learning parameters, which we believe to be highly practical.
>
> Furthermore, since MHA can be easily integrated into many Transformer-based model, it has great potential to contribute to SOTA improvements in various tasks and models in the future.
>
> **[Related Works] Expansion of Related Research**
>
> Thank you for your valuable comments. In addition to [35], we will add a summary and positioning of related research, including further developments of Hopfield-based layers (e.g., [39]), to the camera-ready version. In particular, we plan to discuss this topic in greater detail from the perspective of the relationship between the dynamic state of Hopfield-type models and Transformers.

---

> > ### Comment · Reviewer_7EBm · 2025-08-05
> > **Response to rebuttal of authors**
> >
> > Thank you for providing the additional explanation. If it will be possible to blend this description into the text where appropriate, it may help clarify for other readers of the paper. I have already indicated an accept from my side.

---

> > > ### Author Response · Authors · 2025-08-08
> > >
> > > Thank you very much for your positive evaluation and encouraging comments.
> > >
> > > We will include appropriate supplementary explanations into the manuscript so that readers can clearly understand the context of related research and the practical potential of our proposal. We sincerely appreciate your support, feedback and understanding.

---

### Note · Authors · 2025-08-13

We sincerely thank all reviewers for their constructive and encouraging feedback. Positive remarks such as “Hope to see this paper accepted at NeurIPS. Good luck!” (aW1E) and “I think the work presents good results” (BWoL) reflect the supportive discussions that helped us clarify our contributions.

The main contribution, as understood by reviewers (“The contribution is an elegant theoretical contribution” – 7EBm; “The authors provide an understanding on why the introduced method has a better performance and highlighted the problem of rank collapse in Transformer — cool!” – BWoL), is to **deepen the understanding** of the role of hidden states in Hopfield networks and to **explain why** the proposed method improves performance, **from the perspective of the rank collapse** issue in Transformers.

One concern raised was that “The improvements seem relatively modest compared to the sophistication of the theory” (WaSH). Through the discussion, we clarified that although **we did not aim to maximize performance through extensive engineering**, the results still demonstrate non-trivial gains. While the paper’s focus is not on proposing a practical method, we agree that enhancing practical applicability is valuable. We therefore provided additional results, including LLaMA experiments, which addressed the reviewers’ concerns.  As requested by BWoL (“I would love for the authors to incorporate the results on text datasets”), the camera-ready version will include experiments with full datasets such as BookCorpus, alongside other refinements suggested by reviewers.

Through the review and discussion process, we believe the novelty, theoretical depth, and significance of this work have been communicated effectively. We sincerely thank the reviewers for their engagement and hope that our work will be considered a meaningful contribution.

---

### Decision · Program_Chairs · 2025-09-17

**Decision:**

Accept (poster)

**Comment:**

The paper utilizes the Krotov-Hopfield formulation of modern Hopfiled networks for rewriting the attention mechanism to include hidden units. While conventional attention arises in the adiabatic limit, this novel reformulation makes it possible to depart from the adiabatic limit, and formulate an alternative attention mechanism which benefits from the hidden unit dynamics. The paper argues that this reformulation mitigates the rank collapse problem common in deep transformers.

The idea is novel and the work received enthusiastic evaluations from reviewers. The reviewers request that the additional results with language models, provided during rebuttal, are included in the camera-ready version.